# Overcoming Brittleness in Pareto-Optimal Learning-Augmented Algorithms

**Alex Elenter**
Sorbonne University, CNRS, LIP6
4 place Jussieu
Paris, France 75005
alexelenter@gmail.com

**Spyros Angelopoulos**
International Laboratory on Learning Systems
Montreal, Canada, and
Sorbonne University, CNRS, LIP6
Paris, France 75005
spyros.angelopoulos@lip6.fr

**Christoph Dürr**
Sorbonne University, CNRS, LIP6
4 place Jussieu
Paris, France 75005
christoph.durr@lip6.fr

**Yanni Lefki**[*]
Institut Polytechnique de Paris
Rte de Saclay
Palaiseau, 91120, France
yanni.lefki@gmail.com

## Abstract

The study of online algorithms with machine-learned predictions has gained considerable prominence in recent years. One of the common objectives in the design and analysis of such algorithms is to attain (Pareto) optimal tradeoffs between the *consistency* of the algorithm, i.e., its performance assuming perfect predictions, and its *robustness*, i.e., the performance of the algorithm under adversarial predictions. In this work, we demonstrate that this optimization criterion can be extremely brittle, in that the performance of Pareto-optimal algorithms may degrade dramatically even in the presence of imperceptive prediction error. To remedy this drawback, we propose a new framework in which the smoothness in the performance of the algorithm is enforced by means of a *user-specified profile*. This allows us to regulate the performance of the algorithm as a function of the prediction error, while simultaneously maintaining the analytical notion of consistency/robustness tradeoffs, adapted to the profile setting. We apply this new approach to a well-studied online problem, namely the *one-way trading* problem. For this problem, we further address another limitation of the state-of-the-art Pareto-optimal algorithms, namely the fact that they are tailored to worst-case, and extremely pessimistic inputs. We propose a new Pareto-optimal algorithm that leverages any deviation from the worst-case input to its benefit, and introduce a new metric that allows us to compare any two Pareto-optimal algorithms via a *dominance* relation.

## 1 Introduction

The field of learning-augmented online algorithms has witnessed remarkable growth in recent years, starting with the seminal works of Lykouris and Vassilvitskii [31] and Purohit *et al.* [36]. The focus, in this field, is on improving the algorithmic performance by leveraging some inherently imperfect *prediction* on the online input. This is in contrast to the standard framework of *competitive analysis* [15], in which the algorithm has no access to any information about the future, and the analysis is based on adversarial inputs tailored to the myopic nature of the algorithm.

---

[*]Research done while at LIP6, Sorboonne University.

38th Conference on Neural Information Processing Systems (NeurIPS 2024).

Learning-augmented online algorithms are typically analyzed with respect to three performance metrics. The first is the *consistency* of the algorithm, namely its competitive ratio assuming that the prediction is error-free. The second is the *robustness*, that is, the competitive ratio assuming that the prediction is adversarial, and is thus generated by a malicious oracle. A third consideration is the degradation of the competitive ratio as a function of the prediction error; here, the notion of *smoothness* captures the requirement that the competitive ratio smoothly interpolates between the two extremes, namely the consistency and the robustness.

As expected, not all three objectives can be simultaneously optimized. Many works have thus focused on the trade-off between consistency and robustness. Algorithms with optimal tradeoffs are often called *Pareto-optimal* since their performance lies on the Pareto front of the two extreme metrics. Examples of problems studied in the Pareto setting include online conversion problems [38, 29], searching for a hidden target [4], ski rental [39, 6], online covering [14] metrical task systems [17], energy-minimization scheduling [28], scheduling [7, 5] and online state exploration [23].

Pareto-based analysis is attractive for several reasons. First, it fully characterizes the performance of the algorithm on the extreme scenarios, with respect to the reliability of the prediction. In addition, it provides a mathematically clean formulation of the desired objectives, which is often quite challenging even for seemingly simple online problems. However, as we will discuss, this type of analysis may very well suffer from *brittleness*, in that the performance ratio of any Pareto-optimal algorithm may be as high as its robustness, even if the prediction is near-perfect. This has an important implication for the algorithm designer: namely, in many realistic situations, a Pareto-optimal algorithm may perform even worse than the best competitive algorithm with no predictions.

To illustrate this drawback, as well as our proposed methodology for counteracting it, we will use the well-known *one-way trading* problem, which is one of the fundamental formulations for online financial transactions. In this problem, a decision maker must convert a unit in a given currency, say USD, to a different currency, say EUR, by performing exchanges over an unknown horizon. Specifically, prior to each transaction, the algorithm is informed about the current exchange rate, and must irrevocably exchange a fraction of its USD budget to EUR, according to the rate in question. This problem has served as a proving ground for the competitive analysis of more involved settings such as two-way trading and portfolio optimization; see Chapter 14 in [15] and the survey [35]. In addition, it has connections to other problems such as fractional knapsack [16] and sponsored auctions [41]. Optimal competitive ratios, in the standard framework, were first obtained in [19]. An elegant Pareto-optimal algorithm for maximum-rate prediction was given in [38], based on the concept of an online *threshold* function. However, [38] does not take into consideration the prediction error other than at the two extreme values. In contrast, the interplay between the prediction error across the entire *spectrum* and the performance of the algorithm is at the heart of our study.

## 1.1 Contribution

Our first result (Theorem 3.1) establishes the brittleness of all Pareto-optimal algorithms for one-way trading. To remedy this undesirable situation, in Section 3 we introduce the novel concept of a performance *profile* $F$, chosen by the end user. Informally, $F$ maps the prediction error to an upper bound on the desired performance ratio of the algorithm. This concept is motivated by practical considerations in everyday applications. E.g., in financial markets, a trader may choose a customized profile based on historical stock exchange data, and how accurate past predictions have proven.

Naturally, not every profile may be *feasible*, in that there may not exist an online algorithm whose performance abides with it. Our next main result is an algorithm that decides whether a given profile is feasible (Theorem 3.2). Note that this is an *offline* problem, however, our algorithm also yields an online strategy, if $F$ is indeed feasible. This further allows us to obtain an online algorithm that not only abides with a feasible profile $F$, but also with the "best" possible profile that has a shape similar to that of $F$ (Remark 4.1). We formalize this intuitive notion based on the concept of the best vertical translation of $F$. We thus obtain a generalization of the concept of consistency (which is brittle) to the *consistency according to profile* $F$, which is inherently non-brittle by virtue of the profile definition.

In Section 5, we address another limitation of the known Pareto-optimal algorithms for one-way trading. Specifically, we note that the algorithm of [38] is tailored to worst-case inputs in which the exchange rates increase continuously until a certain point, then drop to the lowest rate. Again from a practical standpoint, such a worst-case scenario never arises in real markets. Motivated by

the concept of the *lenient adversary* of [19] (in the standard, no-prediction setting), we present and analyze an *adaptive*, Pareto-optimal algorithm that leverages any deviation from the worst-case sequence to its benefit. To formally quantify the performance gain, we introduce an additional metric that captures the profit of the algorithm on all exchange rates that are at least as high as the predicted maximum rate, and allows us to compare any two Pareto-optimal algorithms via a *dominance* relation. Another novelty of our algorithm, in the context of the problem at hand, is that it does not require the prediction to be given ahead of time, instead the prediction can be revealed during its execution (Remark 5.1). This is a clearly desirable algorithmic feature, that has been achieved in other online problems, e.g., [9].

In Section 6 we give an experimental evaluation of all our algorithms, over both real data (Bitcoin exchange rates) and synthetic data, which validates the theoretical results and quantifies the obtained performance improvements. We emphasize that our framework can be readily applicable to other learning-augmented problems, in particular those which suffer from brittleness. We discuss another well-known application from AI, namely *contract scheduling* [7] in Section 7.

In terms of techniques, our algorithms and analysis are based on the concept of a *threshold* function which carefully guides the actions of the algorithm. While online threshold algorithms have been used in previous studies, including one-way trading [41, 40, 38, 29], the settings we study pose novel challenges. For the profile-based setting, the design of the function must take into consideration all the constraints induced by the profile. To this end, we use an iterative approach that considers the constraints incrementally, until they are all satisfied. For the adaptive setting, the threshold function must change dynamically, according to the revealed sequence. This is unlike the standard Pareto setting, in which a static function suffices.

While our framework is directly applicable to single-valued predictions, it can also be applied to more complex settings in which the prediction is a vector of values. This is because the concept of the profile still applies, since the error is defined by a distance norm between the predicted and the actual vector.

## 1.2 Related Work

There has been a significant body of recent work on online algorithms with predictions, see, e.g., the surveys [34, 33]. Several problems have been studied in learning-augmented settings, e.g., paging [31, 22], metrical task systems [9, 17], rent-or buy problems [36, 6, 20, 39, 3], packing and covering [14, 8, 21], scheduling [26, 28, 11, 32, 18, 25], matching [27, 10, 24], graph optimization [1, 2, 12, 13], and many others. This is only a partial list; for a comprehensive summary of the existing literature, we refer the reader to the online repository [30]. As discussed earlier, many works have focused exclusively on consistency/robustness tradeoffs, without an explicit error-based analysis, e.g. [38, 29, 4, 39, 6, 14, 17, 28, 7, 23, 1]. Incorporating smoothness in regards to the prediction error is a challenging task, both in terms of modeling and analysis. For instance, [13, 1] studied online combinatorial optimization problems in which the performance of the online degrades as a function of a distance measure between the predicted and the actual solution. Our work differs from such studies in that the dependency on the prediction error is *user specific*, and can change according to the application setting, while still maintaining the concepts of consistency and robustness.

## 2 Preliminaries

In the one-way trading problem, the input $\sigma$ is a sequence of *exchange rates*, where $p_i$ denotes the $i$-th rate in the sequence. The trader has a starting budget equal to 1. We follow the standard assumption that $p_i \in [1, M]$, where $M$ represents an upper bound on the rates that is known in advance. Once $p_i$ is revealed, the trader must decide the amount to be exchanged to the secondary currency, which cannot exceed her current budget. We consider the general setting in which the horizon $n$ is not known ahead of time. The problem formulation also assumes that the trader is notified once the last rate is revealed, and is thus obliged to exchange all of its remaining fund at rate $p_n$.

An algorithm $A$ decides the fractional exchanges upon revealing of $p_i$, as a function of the previous $i-1$ rates , i.e., the sequence $\sigma[1, i-1]$. We denote by $A(\sigma)$ the *profit* of $A$ on $\sigma$, i.e., the total amount that $A$ has produced after the last exchange. We denote by $p^*_\sigma = \max_{i \in [1,n]} p_i$ the *maximum* rate in $\sigma$ and by $\mathrm{OPT}(\sigma)$ the profit of the optimal offline strategy, hence $\mathrm{OPT}(\sigma) = p^*_\sigma$. The competitive

ratio of $A$ is thus defined as $\text{CR}(A) = \sup_\sigma \frac{\text{OPT}(\sigma)}{A(\sigma)}$. For given $\sigma$, we refer to the ratio $\text{OPT}(\sigma)/A(\sigma)$ as the *performance ratio* of $A$ on $\sigma$. The optimal competitive ratio, denoted by $r^*$ is $\Theta(\log M)$, and more precisely, it is equal to the root of the equation $r^* = \ln \frac{M-1}{r^*-1}$ [19].

Given algorithm $A$, we denote by $w_{A,i}(\sigma)$ and $s_{A,i}(\sigma)$, the *budget* used by $A$ and its accrued *profit* right before $p_i$ is revealed, respectively. We refer to $w_{A,i}(\sigma)$ as the *utilization* of $A$. Formally, for every sequence $\sigma$, and every algorithm $A$, we have $w_{A,i} = w_{A_{i-1}} + x_i$, where $x_i$ is the amount traded on the $i$-th rate, that is, $w_{A,i}$ is the total amount exchanged up to and including the $i$-th request. We also have that $s_i = \sum_{j=1}^{i-1} p_j(w_{j+1} - w_j)$, with $s_1 = 0$. For simplicity, we may omit the input $\sigma$, or the algorithm $A$ when it is clear from context. For example, we will denote by $p^*$ the maximum rate in $\sigma$.

The above definitions assume the standard setting in which the algorithm has no information on the input. In regards to learning-augmented settings, we consider the model of [38] in which the algorithm has an imperfect prediction $\hat{p}$ on $p^*$. We define formally, the *consistency* and the robustness of an algorithm $A$ as $c(A) = \sup_{\sigma: p^*_\sigma = \hat{p}} \frac{p^*_\sigma}{A(\sigma)}$ and $r(A) = \sup_\sigma \sup_{\hat{p} \in [1,M]} \frac{p^*_\sigma}{A(\sigma)}$, respectively. An algorithm $A$ with prediction $\hat{p}$ is *Pareto-optimal* if, for any given $r$, it has robustness at most $r$, and has the smallest possible consistency, which we will denote by $c(r)$.

**Remark 2.1.** It suffices to consider only sequences in which the exchange rates increase up to a certain point, then drop to 1 [19]. Moreover, for any competitively optimal algorithm, the worst-case inputs are such in which the exchange rates increase continuously, i.e., by infinitesimal amounts.

## 3 Brittleness of Pareto-Optimal Algorithms and Performance Profiles

We first define formally the concept of *brittleness*.

**Definition 3.1.** Let $\hat{p}$ denote a maximum-rate prediction for $p^*_\sigma$. We say that $\hat{p}$ is $brittle$ if for any Pareto-optimal strategy $A$ of robustness $r$ and consistency $c(r)$, and for every $\epsilon > 0$, there exists $\sigma$ with $|\hat{p} - p^*_\sigma| \leq \epsilon$, for which $\frac{p^*(\sigma)}{A(\sigma)} = r$.

The definition deems a prediction to be brittle if there exist sequences for which the slightest prediction error forces every Pareto-Optimal strategy to have a performance that is equal to its robustness.

**Theorem 3.1** (Appendix A). The maximum-rate prediction is brittle for one-way trading.

Theorem 3.1 shows that Pareto-optimality is a very "fragile" metric for comparing strategies with max-rate prediction. To remedy this drawback, we introduce the new concept of a *profile*.

**Definition 3.2.** Let $\mathcal{P}$ be a partition of $[1, M]$ to $l$ intervals, i.e., $\mathcal{P} = \bigcup_{i=1}^{l}[q_i, q_{i+1})$, with $q_1 = 1$ and $q_{l+1} = M$, and let $\hat{p}$ be a maximum-rate prediction. A *profile* function $F : \mathcal{P} \to \mathbb{R}^+$ is a step function that maps each interval in $\mathcal{P}$ to $t_i \in \mathbb{R}^+$, and which satisfies the following conditions. There exists $\hat{i} \in [1, l]$ such that: (i) $t_{i-1} \geq t_i$, for all $i \leq \hat{i}$ and $t_{i+1} \geq t_i$, for all $i \geq \hat{i}$, and (ii) $\hat{p} \in [q_{\hat{i}}, q_{\hat{i}+1})$.

The profile function allows the end user to impose a requirement on the performance of the algorithm, as expressed in the following definition.

**Definition 3.3.** We say that an online strategy $A$ *respects* a given profile $F : \bigcup_{i=1}^{l}[q_i, q_{i+1}) \to \mathbb{R}^+$ if for all input sequences $\sigma$ for which $p^*_\sigma \in [q_i, q_{i+1})$, it holds that $\frac{\text{OPT}(\sigma)}{A(\sigma)} \leq F([q_i, q_{i+1}))$.

Informally, a profile $F$ reflects a desired worst-case performance of an algorithm, assuming that the *actual* but unknown maximum rate in the input sequence is in the interval $[q_i, q_{i+1})$. Thus, the profile represents the desired upper bound on the performance of an algorithm, as a function of the prediction error. Unlike Pareto-optimality, which only cares about performance at extremes, the relation between performance and prediction error becomes now definable across the entire *spectrum* of error. The definition also reflects the expectation that the algorithm performs best when the prediction is error-free, and its performance degrades monotonically as a function of the error.

We illustrate the above concepts using the profile depicted in Figure 1a. Here, the profile consists of $l = 6$ intervals, where the first 3 intervals correspond to the *decreasing* part of the profile and the

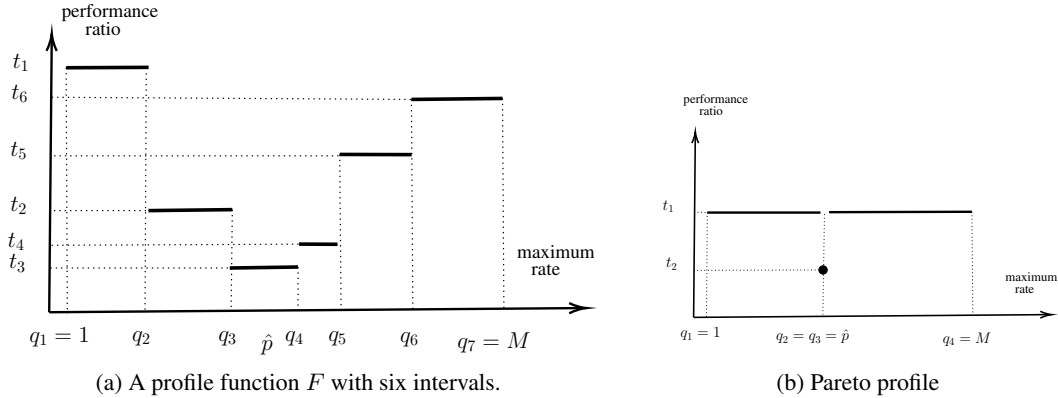

(a) A profile function $F$ with six intervals.     (b) Pareto profile

Figure 1: Illustration of profile functions.

last 4 to the *increasing* part of the profile. Note that the interval $[q_3, q_4)$ contains the prediction $\hat{p}$ and belongs in both the decreasing and the increasing parts. Note also that the profile allows to define an asymmetric dependency on the prediction error. This is a very useful property in applications such as one-way trading. For example, a trader may want to be more cautious if the market will perform worse in the future, than better in the future, relative to what has been predicted.

Figure 1b depicts a different profile in which the performance ratio must be at most $t_1$, for any error, unless the prediction is error-free, in which case the performance ratio has to be at most $t_2 < t_1$. Such a profile yields Pareto-optimality, if $t_1 = r$ and $t_2 = c(r)$.

We are interested in profiles $F$ that are *feasible*, in the sense there exists an online algorithm that respects $F$. The following is one of our main results, whose proof will follow from Theorem 4.1 and Corollary 4.1, as we will show in Section 4.

**Theorem 3.2.** Given a profile $F$ defined over $l$ intervals, there exists an algorithm for deciding whether $F$ is feasible that runs in time $O(l)$. Furthermore, if $F$ is feasible, there exists an *online* algorithm that respects $F$.

Given our algorithm that decides the feasibility of a profile, we can also answer a more general question. Suppose that $F$ is infeasible, but we would like, nevertheless, to be able to respect a profile $F'$ that is "similar" to $F$. Conversely, if $F$ is feasible, then we know we can likely do even better, for example, we would like to follow a profile $F'$ that is similar to $F$, but maps some intervals to smaller ratios. The following definition formalizes this intuitive objective.

**Definition 3.4.** Let $F : \mathcal{P} \to \mathbb{R}^+$ denote a profile. Given $a \in \mathbb{R}^+$, we define the *extension* $G_a$ of $F$ as the vertical transformation of $F$, in which, for every interval $[q_i, q_{i+1}) \in \mathcal{P}$ it holds that $G_a([q_i, q_{i+1})) = a \cdot F([q_i, q_{i+1}))$.

We can generalize the concepts of consistency and robustness *relative to a profile* $F$ as follows, recalling that $\hat{p} \in [q_{\hat{i}}, q_{\hat{i}+1})$.

**Definition 3.5.** Given a profile $F$ for a prediction $\hat{p}$, and a robustness $r$, we say that algorithm $A$ is $r$-robust and $c$-consistent *according to* $F$, if there exists an extension $G_a$ of $F$ for which: (i) for every interval, we have $G_a([q_i, q_{i+1})) \leq r$; (ii) $G_a([q_{\hat{i}}, q_{\hat{i}+1})) \leq c$; and (iii) $A$ respects $G_a$.

**Remark 3.1.** The smoothness of a profile is related to the number of intervals, $l$. The larger the $l$, the smoother the performance of an algorithm which respects the profile.

## 4   Profile-Based Algorithms

In this section, we present an algorithm which decides whether a given profile $F$ is feasible or not. Note that this is an *offline*, decision problem, which we will denote by FEASIBLE($F$). In addition, if $F$ is feasible, we also provide an *online* algorithm that respects $F$.

Our algorithms are inspired by the class of *threshold* algorithms (OTA), introduced in [41]. In these algorithms, a threshold function $\Phi$ guides the decision about the amount to be exchanged when a

---

**Algorithm 1** Algorithm PROFILE for FEASIBLE (F); also an online strategy if $F$ is feasible

---

**Input:** $F : \mathcal{P} = \bigcup_{i=1}^{l} [q_i, q_{i+1}) \to \mathbb{R}^+$. Denote $F([q_i, q_{i+1}))$ by $t_i$.

1: $w_1 \leftarrow 0,\ s \leftarrow 0$
2: **for** $i \in 1, \ldots, l$ **do**
3: $\quad \rho_i \leftarrow t_i \cdot (s + 1 - w_i)$
4: $\quad$ **if** $\rho_i \geq q_i$ **then**
5: $\quad\quad w_{i+1} \leftarrow \frac{1}{t_i} \cdot \ln\left( \frac{q_{i+1} - 1}{\rho_i - 1} \right) + w_i$
6: $\quad\quad \Phi_i(w) \leftarrow \begin{cases} \Phi_{i-1}(w) & \text{if } w \in [1, w_i) \\ (\rho_i - 1) \cdot e^{t_i \cdot (w - w_i)} + 1 & \text{if } w \in [w_i, w_{i+1}) \end{cases}$
7: $\quad\quad s \leftarrow s + \int_{w_i}^{w_{i+1}} \Phi_i(t)\, dt$
8: $\quad$ **else**
9: $\quad\quad w_i' \leftarrow \frac{q_i - t_i \cdot (s_i - w_i q_i + 1)}{t_j \cdot (q_i - 1)}$
10: $\quad\quad s' \leftarrow s + q_i \cdot (w_i' - w_i)$
11: $\quad\quad w_{i+1} \leftarrow \frac{1}{t_i} \cdot \ln\left( \frac{q_{i+1} - 1}{t_i \cdot (s' + 1 - w_i') - 1} \right) + w_i'$
12: $\quad\quad \Phi_i(w) \leftarrow \begin{cases} \Phi_{i-1}(w) & \text{if } w \in [1, w_i) \\ (t_i \cdot (s' + 1 - w_i') - 1) \cdot e^{t_i \cdot (w - w_i')} + 1 & \text{if } w \in [w_i', w_{i+1}) \end{cases}$
13: $\quad\quad w_i \leftarrow w_i'$
14: $\quad\quad s \leftarrow s' + \int_{w_i}^{w_{i+1}} \Phi_i(t)\, dt$
15: **if** $w_{l+1} > 1$ **then return** $F$ is infeasible **else return** $F$ is feasible and output $\Phi_l$

---

new rate is revealed. Specifically, $\Phi$ maps *utilization* to *reservation rates*. Here, a utilization value $w \in [0, 1]$ represents the fractional amount exchanged so far by the online algorithm, whereas the reservation rate, $\rho$, is the minimum rate in $[1, M]$ at which the algorithm will make an exchange. At each point a new rate $p_i$ is revealed, the algorithm updates its utilization by setting $w_{i+1} = \Phi^{-1}(p_i)$, if $p_i > \Phi(w_i)$, otherwise $w_{i+1} = w_i$. In both cases, it exchanges an amount equal to $w_{i+1} - w_i$ at rate $p_i$. The function $\Phi$ must be increasing, and its codomain must include $[1, M]$.

The main challenge posed in our setting is to guarantee the varying performance ratios globally, i.e., for all intervals and not just locally for a given interval. Thus, we need a global approach that takes into account the entirety of the profile, and in particular the transitions between consecutive intervals. We will thus design a function $\Phi$ so as to satisfy $l$ set of constraints, where each set of constraints applies to a specific interval. Define $\tilde{s}_i = \int_0^{w_i} \Phi(u)du$, with $\tilde{s}_1 = 0$. We seek a function $\Phi$ and values $0 = w_1 \leq \ldots \leq w_{l+1} \leq 1$ such that the following constraints are satisfied for all $i \in [1, l]$.

$$[\beta] \qquad\qquad \forall \beta \in [w_i, w_{i+1}) : \frac{\Phi(\beta)}{\tilde{s}_i + \int_{w_i}^{\beta} \Phi(t)\, dt + 1 - \beta} \leq t_i.$$

$$[w_{i+1}] \qquad\qquad \Phi(w_{i+1}) = q_{i+1}.$$

$$[u] \qquad\qquad w_i \leq w_{i+1} \leq 1.$$

Constraint $[\beta]$ expresses the requirement on the performance ratio $F([q_i, q_{i+1}))$ that is imposed by the profile. Note that here $\tilde{s}_i$ is the minimum profit of an OTA at the point it reaches utilization $w_i$. This follows from Remark 2.1. Constraint $[w_{i+1}]$ allows us to obtain the partition of the utilization levels induced by the profile. Moreover, such a constraint is needed for constraint $[\beta]$ to correctly represent the performance ratio indicated by the profile. Constraint $[u]$ establishes that the utilization levels defined are increasing and that they do not exceed the unit budget available to the algorithm. The following lemma follows straightforwardly from the above discussion.

**Lemma 4.1.** $F$ is feasible if and only if there exist $\Phi$ and $w_1, \ldots, w_{l+1}$ that satisfy the above sets of constraints, for all $i \in [1, l]$.

Algorithm 1, which we call PROFILE, shows how to obtain the threshold function $\Phi$, along with the utilization values $w_1, \ldots w_{l+1}$, assuming that $F$ is feasible. This is formally stated in Theorem 4.1. We emphasize that the theorem proves an even stronger statement; namely, if $F$ is not feasible, then PROFILE correctly outputs its infeasibility. That is, the algorithm fully solves FEASIBLE($F$).

**Theorem 4.1** (Appendix B). A profile $F$ admits an online strategy which respects $F$ if and only if PROFILE terminates with a value $w_{l+1} \leq 1$.

Furthermore, if $F$ is feasible, then PROFILE directly provides an online algorithm that respects $F$:

**Corollary 4.1.** If $F$ is feasible, then the threshold function $\Phi_l$ returned by PROFILE defines an OTA which respects $F$.

**Remark 4.1.** For a profile $F$, we can use binary search in combination with PROFILE, in order to find the minimum $a \in \mathbb{R}^+$, such that $G_a$ extends $F$ and $G_a$ is feasible, according to Definition 3.4.

We give some intuition about PROFILE, and how we obtain $\Phi$, and the values $w_i$, for all $i$. The algorithm computes $\Phi$ incrementally: namely, in iteration $i$, it obtains a new function $\Phi_i$ that aims to satisfy the sets of constraints for the intervals $\bigcup_{k=1}^{i} [q_k, q_{k+1})$, and computes a value for $w_{i+1}$, as well as an updated value for $w_i$. In each iteration $i$, the algorithm guarantees that an OTA based on $\Phi_i$ respects the profile on all sequences whose maximum rate is in $[1, q_{i+1})$ (provided that this is indeed feasible) and, furthermore, that the utilization at the end of iteration $i$, namely $w_{i+1}$ is as small as possible. This is crucial, since it allows us to decide FEASIBLE($F$) based on the final value of $w_{l+1}$.

The algorithm makes a distinction between two types of updates. The first type occurs in the increasing part of the profile, i.e., when $t_i < t_{i-1}$. This is a relatively simpler case, because the algorithm has already guaranteed a smaller ratio in the previous interval. Hence the algorithm can afford to wait until it sees a rate that exceeds the reservation rate $\rho_i$ (line 5-7). The second type occurs in the decreasing part of the profile (lines 9-14). This is intuitively a harder case, because on every new interval the algorithm must do even better than in the previous intervals. That is, when observing a rate equal to $q_i$, the algorithm now needs to perform at a ratio $t_i < t_{i-1}$, hence it should have made a bigger profit. To this end, we need first to increase $w_i$ (lines 9 and 13) then extend $\Phi_{i-1}$ to account for interval $i$ (line 12). The precise amount by which we increase $w_i$ is guided by the requirement that the algorithm must have performance ratio $t_i$ for the worst-case sequence of increasing rates up to $q_i$.

## 5 An Adaptive Pareto-Optimal Algorithm

In this section we study another generalization of Pareto-optimality. The starting observation is that the Pareto-optimal OTA of [38] is tailored to worst-case scenarios. Namely, the threshold function in [38] is *static*, i.e., determined prior to the execution of the algorithm, and tailored to a sequence of continuously increasing exchange rates that may suddenly drop to 1. However, in practice, such sequences never occur in real markets. We show how to obtain an algorithm that is not only Pareto-optimal, but also leverages deviations from the worst-case sequence to its benefit.

Our setting is further motivated by [19], who studied the basic setting of standard competitive analysis without predictions. Their solution is based on *threat-based* policies, i.e., algorithms that exchange at each point in time the minimum required amount so as to guarantee the optimal competitive ratio. In this section, instead, we consider the learning-augmented setting in which the algorithm has access to a max-rate prediction $\hat{p}$. Our algorithm uses an *adaptive* threshold policy, in which the threshold function is updated every time a deviation from the worst-case input is observed. We follow this approach since OTAs are typically more versatile than threat-based policies, and can apply to more complex problems and settings, such as several variants of the knapsack problem, e.g., [40].

In a nutshell, we seek a Pareto-optimal algorithm that is not only optimal over worst-case sequences, but also over all other sequences. To describe this formally, we first define some concepts. Let $\hat{p}$ be a max-rate prediction for an input $\sigma$ of increasing rates, and define $\tilde{\sigma}$ as the suffix of $\sigma$ comprised of rates at least as high as $\hat{p}$. (in the event that $\tilde{\sigma}$ is the empty sequence, our problem reduces to standard Pareto optimality). Let $\tilde{\sigma} = \tilde{p}_1, \ldots, \tilde{p}_m$, and $\tilde{s}_{i+1}(A, \sigma)$ denote the profit made by an online algorithm $A$ on $\sigma$ after its exchange over rate $\tilde{p}_i$, for any $i \in [1, m]$, . Let also $\mathbf{s}(A, \sigma)$ denote the vector $\langle \tilde{s}_i(A, \sigma) \rangle : i \in [1, m]$. We say that algorithm $A$ *dominates* another algorithm $B$ on input $\sigma$, if $\mathbf{s}(A, \sigma)$ is lexicographically no smaller than $\mathbf{s}(B, \sigma)$.

Informally, $\mathbf{s}(A, \sigma)$ is the vector of profits that $A$ has made so far, for each rate that is at least as high as the predicted maximum rate. The lexicographic ordering assigns priority to profits made at exchange rates close to, but larger than the prediction. We now state our main result.

**Theorem 5.1** (Appendix C). For any robustness requirement $r$, ADA-PO is Pareto-optimal and dominates every other Pareto-optimal algorithm, on every possible sequence $\sigma$.

Note that the algorithm of [38] is dominant only for sequences in which the exchange rates increase continuously up to some $p^* \geq \hat{p}$, then drop to 1. For those and all other sequences, our algorithm dominates that of [38]. Note also that a dominant $r$-robust algorithm is a Pareto-optimal algorithm.

---

**Algorithm 2** ADA-PO (adaptive Pareto-optimal)

---
**Input:** $r \in \mathbb{R}, \hat{p} \in [1, M]$
1: $w \leftarrow 0, s \leftarrow 0, p^* \leftarrow 1$
2: **for** $p_i \in \sigma$ **do**
3:     **if** $p_i > p^*$ **then**
4:         $p^* \leftarrow p_i$
5:         **if** $p_i \leq \hat{p}$ **then**
6:             $w_{i+1} \leftarrow \frac{p_i - r \cdot (s + 1 - w p_i)}{r \cdot (p_i - 1)}$
7:             $s \leftarrow s + p_i \cdot (w_{i+1} - w)$
8:             $w \leftarrow w_{i+1}$
9:         **else**
10:             **if** $r \cdot (s + 1 - pw + w^*) \geq M$ **then**
11:                 $w_{i+1} \leftarrow 1$
12:             **else**
13:                 $w_{i+1} \leftarrow w^*$
14:             $s \leftarrow s + p_i \cdot (w_{i+1} - w)$
15:             $w \leftarrow w_{i+1}$

---

ADA-PO consists of two phases. The first phase (lines 5-9) consists of revealed rates strictly smaller than $\hat{p}$. In this phase, the algorithm exchanges the minimum amounts so as to guarantee $r$-robustness (i.e., it makes threat-based decisions). Here, adaptivity allows the algorithm to reserve its budget for the second phase. The second phase (lines 11-15) consists of revealed rates at least as high as $\hat{p}$. This is the challenging part, since we need to ensure simultaneously dominance and $r$-robustness, but these two objectives are in a trade-off relation. Here, adaptivity allows us to exchange more money at each revealed rate without sacrificing robustness.

Suppose that $p_i$ is revealed in the second phase (i.e., $p_i \geq \hat{p}$). To achieve simultaneously the robustness and the dominance, we need to find a continuous increasing $\Phi$ whose domain is $[w_{i+1}, 1]$, along with a value for $w_{i+1}$. To this end, we solve the optimization problem $O_i$, described below.

Here, constraint $[\beta]$ is for guaranteeing $r$-robustness; and constraint $[M]$ and $[u]$ guarantee that $\Phi$ is well-defined as a threshold function. Maximizing $w$ maximizes the amount exchanged at rate $p_i$, which is essential for dominance. In Appendix C we give further details, and we show that $O_i$ has optimal solution $w^*$ equal to the root of the equation $w^* = 1 - \frac{1}{r} \ln \left( \frac{M-1}{r(s_i + 1 - p_i w_i + w^*(p_i - 1) - 1)} \right)$, which is used in line 13 of ADA-PO.

$$
\begin{aligned}
\max \quad & w & (O_i)\\
\text{subj. to} \quad & \\
[\beta] \quad & \forall \beta \in [w, 1) : \frac{\Phi(\beta)}{s_i + p_i \cdot (w - w_i) + \int_w^\beta \Phi(t)\, dt + 1 - \beta} = r,\\
[M] \quad & \Phi(1) \geq M,\\
[u] \quad & w_i \leq w \leq 1.
\end{aligned}
$$

**Remark 5.1.** ADA-PO, unlike the known static OTAs, does not require a prediction $\hat{p}$ ahead of time; the prediction can be revealed during its execution instead, since it is only used in the second phase. This can be very useful in practice, e.g., if the trader obtains information "on-the-fly".

## 6 Experimental evaluation

We present experimental results for both the profile-based algorithm PROFILE (Algorithm 1) and the adaptive Pareto-optimal algorithm ADA-PO (Algorithm 2). We compare our algorithms to the state of the art Pareto-optimal algorithm of [38], which we denote by PO.

**Profile setting.** We use a profile $F$ that consists of three intervals $[q_1 = 1, q_2)$, $[q_2, q_3)$ and $[q_3, q_4 = M]$, where $M = 100$. The profile is defined in terms of the prediction $\hat{p}$, by choosing $q_2 = 0.9\hat{p}$ and $q_3 = 1.1\hat{p}$. In addition, $F$ is such that $F([q_1, q_2)) = t_1 = F([q_3, q_4]) = t_3 = r$, where $r = 4$ (larger than, but close to the optimal competitive ratio $r^*$). Here, $F([q_2, q_3)) = t_2 < r$ is the *smallest* value such that $F$ is feasible. To find $t_2$, we use binary search in $[1, r]$ in combination with PROFILE, and note that this depends on $\hat{p}$. $F$ is depicted in Figure 2a. Intuitively, $r$ corresponds to the robustness, whereas $t_2$ is the performance ratio if the input $\sigma$ is such that $p^* \in [0.9\hat{p}, 1.1\hat{p})$, i.e. if $\hat{p}$ is "close" to $p^*$. The length of $[q_2, q_3)$, which is equal to $0.2\hat{p}$, reflects how much the user trusts the prediction.

Figure 2b depicts the performance of PROFILE, and PO with robustness $r$, on the worst case sequences of maximum rate $p^*$, as a function of $p^*$. Recall that such sequence is of the form $1, \ldots, p^*, 1$, with infinitesimal increments up to $p^*$, simulated using a step equal to 0.01. We denote this sequence by $\sigma_{p^*}^w$. We choose $\hat{p}$ u.a.r. in $[1, M]$ ($\hat{p} = 67.8$ in Figure 2b). We observe that PO exhibits high brittleness if $p^*$ is very close, but smaller than $\hat{p}$, namely has performance ratio of $r$, which validates Theorem 3.1. In contrast, PROFILE guarantees a performance ratio equal to $t_2$ in the entire interval $[0, 9\hat{p}, 1.1\hat{p}]$, as required by $F$, thus tolerating a prediction error as high as $10\%$, while remaining $r$-robust for all errors. This validates Theorem 4.1. As expected, PO has better ratio if $p^* = \hat{p}$ (from the definition of Pareto optimality).

To further quantify the performance difference between the two algorithms, we evaluated both algorithms on 100 randomly defined worst-case sequences. Each sequence $\sigma_{p^*}^w$ is obtained by sampling $\hat{p}$ u.a.r. in $[1, M]$, and for such $\hat{p}$, by randomly picking $p^* \in [0.9\hat{p}, 1.1\hat{p}]$, the significant prediction error for the user. Figure 2c depicts the relative performance difference of the two algorithms for each $\sigma_{p^*}^w$, as a function of the prediction error. We observe that if $p^* < \hat{p}$, then PROFILE improves upon PO by $20\%$ to $50\%$, whereas if $p^* > \hat{p}$, PROFILE is inferior by only $10\%$ to $20\%$. The average improvement we report, taken over the 100 ratios is $22\%$. We conclude that while both algorithms guarantee robustness $r$, PROFILE is not only smooth around the prediction, but also performs better on the average, which supports the benefits from using a profile.

In addition, we performed experiments over sequences obtained from real trading data, using the profile $F$ as above. We used exchange rates from Bitcoin (BTC) to USD; specifically, we used a list of the last 1000 daily exchange rates (finishing on May 20, 2024), defining as the prediction $\hat{p}$ the maximum rate in the first 200 rates, and running the algorithm on a sequence consisting of the last 800 rates. Figure 2d depicts the performance ratios of PROFILE and PO, where each point in the plot corresponds to the maximum rate observed so far: these are the only rates at which the algorithms make exchanges. We observe that PO continues to suffer from brittleness, whereas PROFILE still exhibits smooth degradation in the interval $[0.9\hat{p}, 1.1\hat{p}]$.

In Appendix E we report an additional experiment on the average performance over BTC sequences. The key takeaway from all experiments on both synthetic and real sequences is that PROFILE performs much better if $p^* < \hat{p}$, and at the same time it is only slightly worse, if $p^* > \hat{p}$. This behavior is due to the smoothness enforced around the prediction, as guaranteed by the profile.

**Adaptive setting.** Since, by definition, PO and ADA-PO perform the same over worst-case sequences, we focus on sequences from BTC rates. Based on a list of the last 1000 daily BTC rates, we obtain a prediction $\hat{p}$ and the sequence, as in our profile-based experiments above. Figure 2e plots the performance ratio as a function of the currently observed maximum rate in the sequence. For every such rate that exceeds $\hat{p}$, ADA-PO outperforms PO, which validates Theorem 5.1. This comes at an unavoidable increase in brittleness, as expected, and illustrates the tradeoff between smoothness and dominance. We expect ADA-PO to be the algorithm of choice when the prediction is conservative, or when $\hat{p}$ is not given to the trader ahead of time, but is rather revealed at some point in the sequence.

## 7  Discussion

Our profile-based framework can apply to many other problems augmented with ML predictions, and is not specific to one-way trading. To illustrate this, in Appendix D we analyze another application in the context of *contract scheduling*, which is a classic problem from resource-bounded reasoning in AI, and which, likewise, suffers from brittleness. Our work is the first towards understanding the power and limitations of imperfect ML predictions in competitive financial optimization beyond extreme values of the prediction error. The techniques introduced will help address problems such as two-way trading and portfolio optimization, which have not yet been studied in learning augmented

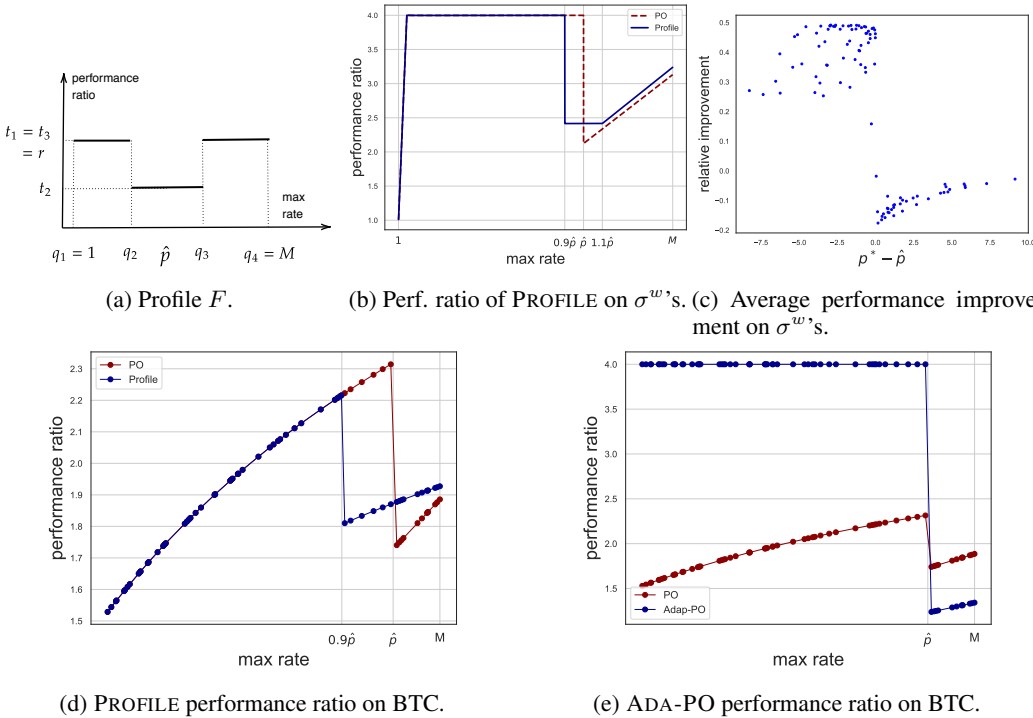

(a) Profile $F$. (b) Perf. ratio of PROFILE on $\sigma^w$'s. (c) Average performance improvement on $\sigma^w$'s.

(d) PROFILE performance ratio on BTC. (e) ADA-PO performance ratio on BTC.

Figure 2: Summary of the experimental results.

settings. Other potential applications include several well-known variants knapsack problems, where online threshold algorithms are commonly used, especially in learning-augmented settings [16, 40]. Last, it would be interesting to study dynamic settings, in which the predictions are obtained as the sequence is revealed to the algorithm.

# 8 Acknowledgements

This work was funded by the project PREDICTIONS, grant ANR-23-CE48-0010 from the French National Research Agency (ANR). The first author acknowledges the support of AANI (Agencia Nacional de Investigacion e Innovacion).

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

# Appendix

## A  Details from Section 3

*Proof of Theorem 3.1.* Let $A$ be a Pareto-optimal algorithm of robustness $r$, and consistency $c(r)$. We will show that for any fixed $\epsilon > 0$, there exists a sequence $\sigma$ and a prediction $\hat{p}$ such that $\eta = |\hat{p} - p_\sigma^*| \leq \epsilon$, and $A$ satisfies Definition 3.1. Since $A$ is Pareto-optimal, there exists a non-empty set of sequences $\Sigma_c$, such that for all $\sigma_c \in \Sigma_c$, if $A$ is given as prediction $p_{\sigma_c}^*$, then

$$\frac{p_{\sigma_c}^*}{A(\sigma_c)} = c(r).$$

As shown in [19] we can assume, without loss of generality, that every $\sigma_c$ is increasing, i.e., it is of the form $\sigma_c = p_1, \ldots, p_k, p_{\sigma_c}^*$ with $p_i > p_j$, for all $i < j$, and $p_{\sigma_c}^* > p_k$. We define $\Sigma$ to be the co-domain of the following function, $f$:

$$f \colon \Sigma_c \to \Sigma \text{ such that } f(\sigma_c) = \begin{cases} \sigma_c & \text{if } |p_{\sigma_c}^* - p_k| \leq \epsilon, \\ p_1, \ldots, p_k, p_{\sigma_c}^* - \epsilon, p_{\sigma_c}^* & \text{otherwise.} \end{cases} \tag{A.1}$$

Given a $\sigma \in \Sigma$, let $n = |\sigma| - 1$, and let $x_n$ be the fraction exchanged by $A$. Since $A$ is $r$-robust, it needs to account for the scenario in which the adversary chooses to drop all rates to 1 after exchanging at the rate $p_n$. Thus, $x_n$ must satisfy

$$\frac{p_n}{s_n + p_n \cdot x_n + 1 - x_n - w_n} \leq r,$$

or equivalently,

$$x_n \geq \frac{p_n - r \cdot (s_n + 1 - w_n)}{r \cdot (p_n - 1)}. \tag{A.2}$$

Define $\omega$ to be the RHS of (A.2) Suppose first, that there exists a sequence $\sigma \in \Sigma$ for which $A$ exchanges $x_n = \omega$. In this case, if $A$ is given a prediction $\hat{p} = p_\sigma^*$, then for the the sequence $\sigma_r = \sigma[1, n]$ we have that $|\hat{p} - p_{\sigma_r}^*| \leq \epsilon$, and:

$$\frac{p_{\sigma_r}^*}{A(\sigma_r)} = \frac{p_n}{s_n + p_n \cdot \omega + 1 - \omega - w_n} = r,$$

and the proof is complete in this case.

It thus remains to consider the case that for all $\sigma \in \Sigma$, $x_n > \omega$. Let $x_{n+1}$ be the amount exchanged by $A$ at rate $p_\sigma^*$. We define an online algorithm $A'$, whose statement is given in Algorithm 3. Intuitively, while the rate is below $p_\sigma^*$, $A'$ makes the same decisions as $A$. If the rate is between $p_\sigma^* - \epsilon$ and $p_\sigma^*$, $A'$ exchanges $\omega$. If the rate is precisely $p_\sigma^*$ $A'$ exchanges $x_n$ plus what $A$ did not exchange on rates which were between $p_\sigma^* - \epsilon$ and $p_\sigma^*$. Finally, $A'$ makes the same decisions as $A$ for all rates that exceed $p_\sigma^*$. We will show that $A'$ has robustness at most $r$ and consistency $c_{A'}$ such that $c_{A'} < c(r)$, which contradicts that $A$ is Pareto-optimal.

We first show that $A'$ is $r$-robust. Let $\sigma'$ be an input sequence and $\hat{p}$ a prediction given to $A'$, we will show that $p_{\sigma'}^* \leq rA(\sigma')$. If $p_{\sigma'}^* < \hat{p} - \epsilon$, then has $A'$ made the same decisions as $A$, hence remains $r$-robust. If $\hat{p} - \epsilon < p_{\sigma'}^* < \hat{p}$, then by definition of $\omega$, $A'$ is guaranteed to be $r$-robust. Last, if $p_{\sigma'}^* \geq \hat{p}$, then $A'$ achieves a strictly better profit than $A$.

It remains to show that $A'$ has consistency strictly smaller than $c(r)$. To this end, it suffices to show that: (i) for all $\sigma_c \in \Sigma_c$ it holds that $\frac{\text{OPT}(\sigma_c)}{A'(\sigma_c)} < c(r)$, and that (ii) for all $\sigma' \notin \Sigma_c$ it holds that $\frac{\text{OPT}(\sigma_c)}{A'(\sigma_c)} < c(r)$, assuming that both $A$ and $A'$ are given a prediction $\hat{p} = p_{\sigma'}^*$.

To show (i), note that for $\sigma' \in \Sigma_c$ it holds that $\frac{\text{OPT}(f(\sigma'))}{A(f(\sigma'))} < c(r)$, due to $A$ exchanging $x_n > \omega$ and $A'$ exchanging $x_n = \omega$. If $f(\sigma') = \sigma'$ (first case in (A.1)) then $\frac{\text{OPT}(\sigma')}{A(\sigma')} < c(r)$. Otherwise, (second case in (A.1)) $A(\sigma') > A(f(\sigma'))$ hence the same result holds. To show (ii), observe that $A'(\sigma') > A(\sigma')$ due to $A$ exchanging $x_n > \omega$ and $A'$ exchanging $x_n = \omega$. Hence, by the definition of $\Sigma_c$, we have

**Algorithm 3** Statement of the online algorithm $A'$

---

**Input:** Algorithm $A$, $\hat{p}$, $\epsilon$
1: $p^* = 1$, $e \leftarrow 0$
2: **for** each rate $p_i$ in the input sequence **do**
3:     **if** $p_i > p^*$ **then**
4:        $p^* \leftarrow p_i$
5:        **if** $p_i < \hat{p} - \epsilon$ **then**
6:           Exchange the same amount as $A$
7:        **else if** $\hat{p} - \epsilon < p_i < \hat{p}$ **then**
8:           Exchange $\omega$
9:           $e \leftarrow e + x_i - \omega$
10:        **else if** $p_i = \hat{p}$ **then**
11:           Exchange $x_n + e$
12:        **else**
13:           Exchange the same amount as $A$

---

$$\frac{\text{OPT}(\sigma_c)}{A'(\sigma_c)} < \frac{\text{OPT}(\sigma_c)}{A(\sigma_c)} < c(r),$$

which concludes the proof. $\qquad\qquad\qquad\qquad\qquad\qquad\qquad\qquad\qquad\qquad\qquad\qquad\square$

# B   Details from Section 4

In this section, we show how to compute the function $\Phi$ used in PROFILE (Algorithm 1), for deciding whether a profile $F$ is feasible. Recall that we seek a function $\Phi$ and values $0 = w_1 \leq \ldots \leq w_{l+1} \leq 1$ that satisfy the following sets of constraints.

$[\beta]$                            $\forall \beta \in [w_i, w_{i+1}) : \dfrac{\Phi(\beta)}{s_i + \int_{w_i}^{\beta} \Phi(t)\,dt + 1 - \beta} \leq t_i$

$[w_{i+1}]$            $\Phi(w_{i+1}) = q_{i+1}$
$[\text{u}]$                $w_i \leq w_{i+1} \leq 1$

for each rate interval $[q_i, q_{i+1})$.

As explained in Section 4, our algorithm builds a function $\Phi$ and values $w_i$ in an iterative way. That is, it processes each set of constraints iteratively, and at each step $j \in [1, l]$ it builds a function $\Phi_j$ and computes values $w_1, \ldots, w_{j+1}$ which satisfy the sets of constraints for all intervals $[q_i, q_{i+1})$ with $i \leq j$. Each function $\Phi_j$ and the new values $w_1, \ldots, w_{j+1}$ are a function of $\Phi_{j-1}$ and the previous values $w_1, \ldots, w_{j+1}$.

We explain an iteration of this process. Suppose that the algorithm is at a step where it has computed $\Phi_{j-1}$ and values $w_1, \ldots, w_j$ as to satisfy the sets of constraints for the intervals $[q_i, q_{i+1})$ with $i < j$. Constraint $[\beta]$ requires us to guarantee a ratio of at least $t_j$ for every sequence whose maximum rate is in $[q_j, q_{j+1})$. We derive a function which achieves a ratio *equal* to $t_j$ for such sequences. The equality is sought, instead of the inequality, in order to minimize utilization. Intuitively, enforcing a ratio smaller than $t_j$ would force the algorithm to exchange more money to achieve a bigger profit. Thus the following constraint

$$\forall \beta \in [w_j, w_{j+1}) : \frac{\Phi(\beta)}{s_j + \int_{w_j}^{\beta} \Phi(t)\,dt + 1 - \beta} = t_j,$$

from which we can obtain the differential equation:

$$\dot{\Phi} = t_j \cdot \Phi - t_j, \tag{B.1}$$

which is a separable first order differential equation. We can hence find the unique solution

$$\Phi(\beta) = C \cdot e^{t_j \cdot \beta} + 1.$$

We then apply constraint $[\beta]$, for an arbitrary $\beta \in [w_j, w_{j+1})$, so to find the value of the constant $C$, which yields

$$\boxed{\Phi(\beta) = (t_j \cdot (s_j + 1 - w_j) - 1) \cdot e^{t_j \cdot (\beta - w_j)} + 1} \tag{B.2}$$

The obtained function is the unique solution to such an equation. We denote $\rho_j = t_j \cdot (s_j + 1 - w_j)$.

We then use constraint $[w_{j+1}]$ to find an expression for $w_{j+1}$:

$$\boxed{w_{j+1} = \frac{1}{t_j} \ln \left( \frac{q_{j+1} - 1}{\rho_j - 1} \right) + w_j} \tag{B.3}$$

Note that $\Phi(w_j) = \rho_j$. There are two cases to be analyzed.

First, if $\rho_j > q_j$, then we can define $\Phi_j$ as follows:

$$\Phi_j(w) = \begin{cases} \Phi_{j-1}(w) & \text{if } w \in [1, w_j) \\ (t_j \cdot (s_j + 1 - w_j) - 1) \cdot e^{t_j \cdot (\beta - w_j)} + 1 & \text{if } w \in [w_j, w_{j+1}), \end{cases}$$

where $w_{j+1}$ is defined in (B.3). We say that we extend the previous $\Phi_{j-1}$. This scenario materializes when the algorithm has achieved a profit $s_j$, which allows it to not exchange while observing rates in $[q_j, \rho_j]$ and still remain $t_j$-competitive. This occurs when $t_j > t_{j-1}$, hence it occurs for the increasing part of the profile.

On the other hand, $\rho_j < q_j$, if $t_j < t_{j-1}$. If this case occurs, the algorithm has not obtained a sufficient profit to be $t_j$-competitive when presented with the sequence which continuously increases from 1 to $q_j$, which is the worst-case sequence as stated in Remark 2.1. As we will show in the proof of Theorem 4.1 $w_j$ is the least utilization that can be spent so to satisfy every set of constraints $[q_k, q_{k+1})$ with $k < j$. To enforce a ratio of $t_j$ and still minimize utilization, the algorithm must exchange a bigger amount when rate $q_j$ is revealed, since exchanging more at a lower rate would lead to a larger utilization. To guarantee a ratio of $t_j$ for the continuous increasing sequence, the algorithm should trade an amount equal to $w'_j - w_j$, where $w'_j$ is obtained from:

$$\frac{q_j}{s_j + q_j \cdot (w'_j - w_j) + 1 - w'_j} = t_j$$

and leads to

$$w'_j = \frac{q_j - t_j \cdot (s_j - w_j q_j + 1)}{t_j \cdot (q_j - 1)}.$$

We now wish to extend function $\Phi_{j-1}$, obtained in the previous iteration, so as to satisfy all constraints for interval $[q_j, q_{j+1})$. Let $s'_j = s_j + q_j \cdot (w'_j - w_j)$, which is the profit obtained by the OTA in the worst case where the maximum rate is $q_j$. We may express this problem by a new set of constraints, which are:

$$[\beta] \qquad \forall \beta \in [w'_j, w_{j+1}) : \frac{\Phi(\beta)}{s'_j + \int_{w'_j}^{\beta} \Phi(t) \, dt + 1 - \beta} \le t_j,$$

$$[w_{j+1}] \qquad \Phi(w_{j+1}) = q_{j+1},$$

$$[u] \qquad w'_j \le w_{j+1} \le 1.$$

Note that this set of constraints is the same as the ones we started with, but $s_j$ was replaced by $s'_j$ and $w_j$ by $w'_j$. Hence, the $\Phi$ and $w_{j+1}$ which satisfy the constraints and minimize $w_{j+1}$ are:

$$\Phi(\beta) = (t_j \cdot (s'_j + 1 - w'_j) - 1) \cdot e^{t_j \cdot (\beta - w'_j)} + 1, \tag{B.4}$$

$$w_{j+1} = \frac{1}{t_j} \ln \left( \frac{q_{j+1} - 1}{t_i \cdot (s' + 1 - w'_i) - 1} \right) + w'_j. \tag{B.5}$$

We can now proceed with the proof for Theorem 4.1.

*Proof of Theorem 4.1.* As stated in Remark 2.1, every online strategy will exchange on rates which are best-seen so far. We can hence state every strategy as an OTA. It suffices then to prove the following: There exists an OTA which respects $F$ if and only if PROFILE terminates with a value $w_{l+1} \leq 1$.

Let $F$ be a performance profile. The if direction follows directly from the design of PROFILE. It suffices to observe that the obtained function $\Phi_l$ can be used as the threshold function for an OTA which respects the profile $F$.

To prove the only if direction, we will prove that every $w_i$ obtained by PROFILE is the least utilization needed to satisfy all sets of constraints for intervals $[q_k, q_{k+1})$ for $k < i$. In other words, we will prove that if A is an OTA, which respects $F$, defined by $\Phi$, and where $w_1', \ldots, w_{l+1}'$ are the respective utilization levels reached by A when observing rates $q_1, \ldots, q_{l+1}$, i.e: $\Phi(w_i') = q_i$ for each $i \in [1, \ldots, l+1]$, then $w_i \leq w_i'$. This statement follows, once again, from the design of PROFILE. By replacing the inequality constraint in $[\beta]$ by an equality, we manage to achieve a ratio which is exactly the one demanded by the profile, hence reserving budget for futures rates. PROFILE obtains a function $\Phi_l$ which enforces, for each $i \in [1, l]$ and for each $q \in [q_i, q_{i+1})$ the equation:

$$\frac{q}{\int_1^{\Phi_l^{-1}(q)} \Phi_l(u) du + 1 - \Phi_l^{-1}(q)} = t_i.$$

We conclude that PROFILE minimizes utilization while satisfying every set of constraints, thus proving the theorem. □

Figure 3 illustrates PROFILE. Here we observe that for the increasing part of the profile, $\Phi_i$ with $i \in [4, 7]$ extends $\Phi_{i-1}$ with an exponential function starting at $w_i$, where $\Phi_i(w_i) > \Phi_{i-1}(w_i)$. Here the vertical "jumps" reflect the less stringent requirement in the increasing part (we can afford to reserve our budget for later). For the decreasing part of the profile, $\Phi_i$ with $i \in [1, 3]$ extends $\Phi_{i-1}$ with an exponential function starting at $w_i' > w_i$ (line 9 in the statement) where $\Phi_i(w_i') = \Phi_{i-1}(w_i)$, which is reflected in the presence of straight lines in Figure 3.

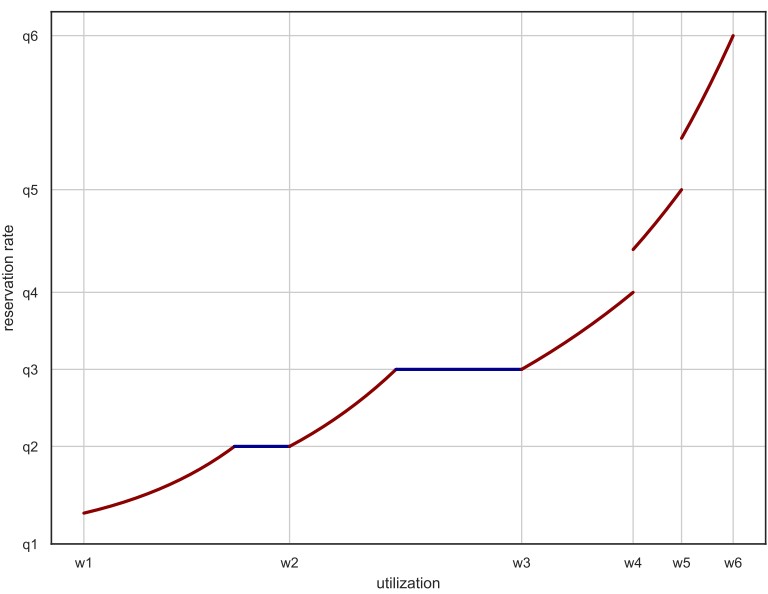

Figure 3: An illustration of PROFILE. Here the profile $F$ is as follows: $F([1, 20]) = 7$, $F([20, 35]) = 5$, $F([35, 50]) = 3$, $F([50, 70]) = 3.5$, and $F([70, 100]) = 4$

.

# C   Details from Section 5

In this section, we detail the calculations that lead to the value $w_{i+1}$, which is the maximum an online algorithm can spend on rate $p_i$ while ensuring $r$-robustness.

The aforementioned $w_{i+1}$ is the solution to the following optimization problem:

$$\max \qquad w \qquad\qquad\qquad\qquad\qquad\qquad\qquad\qquad\qquad (O_i)$$

subj. to

$$[\beta] \qquad \forall \beta \in [w, 1) : \frac{\Phi(\beta)}{s_i + p_i \cdot (w - w_i) + \int_w^\beta \Phi(t)\, dt + 1 - \beta} = r,$$

$$[M] \qquad \Phi(1) \geq M,$$

$$[u] \qquad w_i \leq w \leq 1.$$

From constraint [$\beta$], we do the same analysis as in B to find $\Phi(\beta) = C \cdot e^{r\beta} + 1$. Once again, to find the constant $C$ we use constraint [$\beta$] for an arbitrary value $\beta \in [w_{i+1}, 1]$, which leads to:

$$\Phi(\beta) = \big(r \cdot (s_i + 1 - p_i w_i + w_{i+1} \cdot (p_i - 1)) - 1\big) \cdot e^{r \cdot (\beta - w_{i+1})} + 1.$$

We then use constraint [M] to obtain an upper bound on $w_{i+1}$:

$$\big(r \cdot (s_i + 1 - p_i w_i + w_{i+1} \cdot (p_i - 1)) - 1\big) \cdot e^{r \cdot (1 - w_{i+1})} + 1 \geq M,$$

which leads to:

$$w_{i+1} \leq 1 - \frac{1}{r} \ln \left( \frac{M - 1}{r(s_i + 1 - p_i w_i + w_{i+1}(p_i - 1) - 1)} \right).$$

Thus the largest value of $w_{i+1}$ is the root of the equation

$$w_{i+1} = 1 - \frac{1}{r} \ln \left( \frac{M - 1}{r(s_i + 1 - p_i w_i + w_{i+1}(p_i - 1) - 1)} \right),$$

which can be solved using numerical methods. Let $\rho$ be the reservation rate for utilization $w_{i+1}$, then

$$\rho = \Phi(w_{i+1}) = r \cdot (s_i + 1 - p_i w_i + w_{i+1} \cdot (p_i - 1)).$$

If $\rho > M$, then the algorithm has achieved a sufficient profit to guarantee $r$-robustness independently of future rates. Hence, to maximize $w_{i+1}$, we can safely set it to 1. However, if $\rho < M$, then constraint [M] was saturated, and the algorithm will achieve a performance ratio of $r$ for every sequence which grows continuously from $\rho$ until a rate $p^* \in [\rho, M]$. Moreover, for every sequence whose maximum rate $p^* \in [p_i, \rho)$ the algorithm will have a performance ratio smaller than $r$.

As explained in Appendix B using constraint [$\beta$] with an equality allows us to guarantee a performance ratio of $r$ minimizing utilization. Observe that to maximize $w_{i+1}$ we need to minimize the left-over budget to remain $r$-robust in the future. We can hence conclude that $w_{i+1} - w_i$ is indeed the largest amount of money we can exchange at rate $p_i$ and remain $r$-robust.

We will next provide the proof for Theorem 5.1.

*Proof of Theorem 5.1.* We are to prove that ADA-PO is Pareto-Optimal and dominates every other Pareto-Optimal algorithm on any sequence $\sigma$.

First, we will prove that ADA-PO is Pareto-Optimal. Let $r$ be a robustness requirement, and $c(r)$ the respective consistency. To start with, we prove that ADA-PO is $r$-robust. Consider first the (easy) case where $p^* < \hat{p}$ then ADA-PO assures a performance ratio of $r$ using the threat-based approach.

Consider then the (harder) case in which $p^* > \hat{p}$. Let $p_i$ be the first rate above $\hat{p}$ and $w_{i+1}, \Phi_i$ be the respective solution to problem $O_i$. We must prove that no matter how the sequence continues ADA-PO achieves a performance ratio of at least $r$. If $\Phi(w_{i+1}) \geq M$ then a performance ratio of $r$ is guaranteed, due to $\frac{M}{s_{i+1} + 1 - w_{i+1}} \leq r$, from constraint [$\beta$]. Suppose then $\Phi_i(w_{i+1}) < M$, then by constraints [M] and [u] we know that $w_{i+1} < 1$. When the next rate $p_{i+1} > p_i$ is revealed the same analysis can be applied. We thus obtain a non-decreasing sequence of reservation rates $\Phi_j(p_j)$

for $j > i$. For each rate, problem $O_i$ is solved. Note that the feasibility of problem $O_i$ with rate $p_i$ implies the feasibility of the problem $O_i$ with the next rate as shown by the next analysis. Namely, if $p_i \leq \Phi(p_{i-1})$ then $w = w_i$, $\Phi_i = \Phi_{i-1}$ is a solution, and if $p_i > \Phi(p_{i-1})$, then $w = \Phi_{i-1}^{-1}(p_i)$, $\Phi_i = \Phi_{i-1}$ is as well. Furthermore, both cases lead to a performance ratio of at least $r$ in case the next rate equals 1 and is the last rate. We hence conclude, that either one of the reservation rates is greater or equal than $M$ or ADA-PO successfully achieves a performance ratio of $r$ for each rate ($w_i < 1$ was a solution for each problem). We conclude then that ADA-PO is $r$-robust.

We will now prove that ADA-PO is $c(r)$-consistent. We must prove that for every error-free sequence the performance ratio is at most $c(r)$. Let $A'$ be any Pareto-Optimal algorithm. When observing rates below $\hat{p}$, ADA-PO follows the threat-based policy, hence for every error-free sequence, its budget is at least the same as $A'$ when a rate equal to $\hat{p}$ is exhibited. Then by solving the optimization problem, ADA-PO exchanges the most it can in order to remain $r$-robust, a larger amount would make the problem infeasible. In other words, there would not exist a function $\Phi$ satisfying the constraints, and the continuously increasing function from $\hat{p}$ to $M$ will lead to a performance ratio bigger than $r$. Hence, no other algorithm could achieve a better profit. We conclude that ADA-PO is $c(r)$-consistent.

We finally prove that ADA-PO dominates $A'$. By the previous analysis, when observing the first rate above the prediction, ADA-PO has a budget at least the budget than $A'$. As ADA-PO exchanges the most it can to remain $r$-robust, it will obtain a next utilization which is equal or smaller than $A'$, hence achieving a better profit, because $A'$ exchanged the same or less at lower rates. If $A'$ has behaved the same as ADA-PO, then this process repeats for every following rate. We conclude then that ADA-PO dominates or performs equally to $A'$. $\qquad\square$

**Remark C.1.** To conclude we offer an intuitive explanation of dominance. If the maximum rate of the sequence is below the prediction, then ADA-PO's profit will be smaller or equal than any other Pareto-Optimal algorithm. Its profit will be equal if the sequence is a continuously increasing one. Moreover, for the first rate equal or greater than the prediction, its profit will be greater or equal than any other Pareto-Optimal algorithm. By definition of dominance, while observing rates above the prediction, either the two profits will be equal, or ADA-PO's profit is larger, unless the Pareto-Optimal algorithm attained a smaller profit at an earlier rate.

## D   Profile-based contract scheduling

In this section, we discuss another application of our profile-based framework of Section 3. Specifically, we focus on another well-known optimization problem that has been studied under learning-augmented settings, namely contract scheduling. In its standard variant, the problem consists of finding an increasing sequence $X = (x_i)_{i=0}^{\infty}$ which minimizes the *acceleration ratio*, formally defined as

$$\texttt{acc}(X) = \sup_T \frac{T}{\ell(X, T)}. \tag{D.1}$$

where $\ell(X, T)$ denotes the *largest* contract completed by $T$ in $X$, namely

$$\ell(X, T) = \max_j \{x_j : \sum_{i=0}^{j} x_i \leq T\}.$$

Contract scheduling is a classic problem that has been studied under several settings. In its simplest variant stated above, the optimal acceleration ratio is equal to 4 [37], but many more complex settings have been studied in the literature; see [7] and references therein. In this section we are interested in the learning augmented setting introduced in [7] in which there is a *prediction* $\tau$ on the interruption time $T$. The prediction *error* is defined as $\eta = |T - \tau|$. In this context, the consistency $c(X)$ of schedule $X$ is defined as

$$c(X) = \frac{\tau}{\ell(X, \tau)},$$

whereas its robustness is defined as

$$r(X) = \sup_{T \geq 1} \frac{T}{\ell(X, T)},$$

i.e., the worst-case performance of $X$, assuming adversarial interruptions. Since the latter occur arbitrarily close to the completion time of any contract, we obtain an equivalent interpretation of the robustness as

$$r(X) = \sup_{i \geq 1} \frac{\sum_{j=0}^{i} x_j}{x_{i-1}}.$$

In [7] it was shown that the optimal consistency of a 4-robust schedule is equal to 2. However, as proven in [5], any such schedule suffers from brittleness. Namely, for any $\epsilon > 0$, there exists a prediction $\tau$ and an actual interruption time $T$ such that $|T - \tau| = \epsilon$, and any 4-robust and 2-consistent schedule satisfies $\ell(X, T) \leq \frac{T+\epsilon}{4}$.

In the remainder of this section we will show how to use our framework of profile-based performance so as to remedy this drawback. For definiteness, and to illustrate the application of the techniques, we consider the requirement that the performance of the schedule degrades linearly as a function of the prediction error. Namely, suppose that we require that $f(X, T) := T/\ell(X, T)$ be respect a profile $F_\phi$, where the latter is defined as a symmetric, bilinear function that is decreasing for $T \leq \tau$, and increasing for $T \geq \tau$, with slope $\phi$, as illustrated in Figure 4. This profile is chosen by the schedule designer, and the angle $\phi$ captures the "smoothness" at which the schedule is required to degrade as a function of the prediction error.

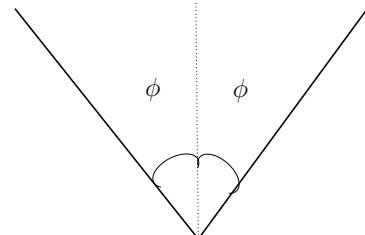

Figure 4: An illustration of the profile $F_\phi$.

More specifically, for a given prediction $\tau$, and a profile $F_\phi$ as above, we are interested in finding the best extension of $F_\phi$ such that there exists a 4-robust schedule that respects the extension. We can thus define the analytical concept of *consistency according to $F_\phi$* as

$$c_{F_\phi} := \sup_\tau \inf_T \frac{T}{\ell(X, T)} \; : \; X \text{ respects } F_\phi.$$

The following theorem states our main result.

**Theorem D.1.** Given a profile $F_\phi$ and a prediction $\tau$ on an interruption time, we can compute a 4-robust schedule that respects $F_\phi$ and has optimal consistency according to $F_\phi$.

*Proof.* We will assume that $X$ if of the form $(\lambda 2^i)_{i \in \mathbb{Z}}$. This is not a limiting assumption, as discussed in [5], and its purpose is to simplify the calculations. Since any 4-robust schedule is of the above form [5], it will suffice to compute a $\lambda$ that satisfies the constraints of our problem, and the result will follow.

Recall that $f(X, T)$ denotes the function $T/\ell(X, T)$. By definition, for every $i \in \mathbb{N}$, $f(X, T)$ is a linear, increasing function of $T$ function in the interval $I_k = [T_k, T_{k+1}] = [\lambda 2^k, \lambda 2^{k+1}]$, with smallest value equal to 2, and largest value equal to 4.

With the above observation in mind, for a given, fixed $\lambda$, let $k$ be such that $\tau \in I_{k+1}$, i.e., we have that $\ell(X, \tau) = \lambda 2^k$. Define $\alpha \in [1, 2]$ to be such that $\tau = \alpha T_k$, and note that by construction, $\alpha$ is a function of $\lambda$. Moreover

$$f(X, \tau) = \frac{\tau}{\lambda 2^k} = \frac{\alpha T_k}{\lambda 2^k} = \frac{\alpha \lambda 2^{k+1}}{\lambda 2^k} = 2\alpha, \tag{D.2}$$

which implies that it suffices to compute $\alpha$, then $\lambda$ must be chosen so that $\lambda = 2^{\{\log(2\alpha)\}}$, where $\{x\}$ denotes the fractional part of $x$.

In order to minimize $f$, subject to $X$ respecting the profile, $\lambda$ must be chosen such that one of the two cases occur, which we analyze separately.

*Case 1.* The profile $F_\phi$ has a unique intersection point with $f$ at $T = \tau$, and moreover $F(T_k + \epsilon) = 4$, for infinitesimally small $\epsilon > 0$. This situation is illustrated in Figure 5. For this case to arise, and for the schedule to be consistent with $F$, it must be that

$$\tan(\frac{\pi}{2} - \phi) \geq \frac{4 - 2}{T_{k+1} - T_k} = \frac{2}{T_k} = \frac{2\alpha}{\tau}. \tag{D.3}$$

It must then be that $f(X, \tau) + \frac{\tau - T_k}{\tan \phi} = 4$, hence

$$4 - \rho(1 - \frac{1}{\alpha}) = 2\alpha, \text{ where } \rho = \frac{\tau}{\tan \phi}.$$

Solving the above equality for $\alpha$ minimizes $f$, by means of (D.2). We obtain that

$$\alpha = \frac{1}{4}(\sqrt{\rho^2 + 16} - \rho + 4) \text{ and } f(X, \tau) = 2\alpha,$$

subject to the condition (D.3).

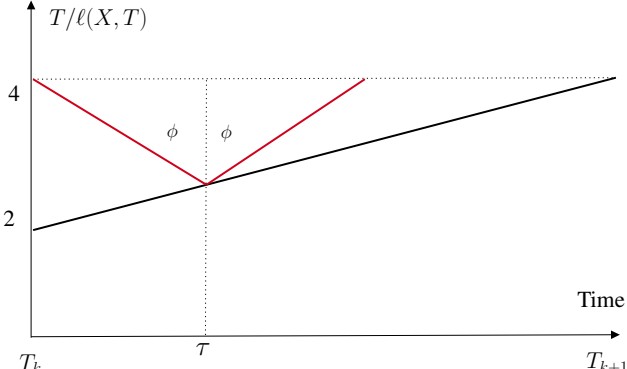

Figure 5: An illustration of Case 1.

*Case 2.* This case occurs if the condition in Case 1 does not apply. The profile $F_\phi$ is such that $F(T_k + \epsilon) = F(T_{k+1} - \epsilon)$, for infinitesimally small $\epsilon > 0$. This situation is illustrated in Figure 6. For this case to arise, and for the schedule to respect $F_\phi$ it must be that $\tau = \frac{T_{k+1} + T_k}{2} = \frac{3}{2}\frac{\tau}{\alpha}$, hence $\alpha = 3/2$. In this case, we obtain that

$$f(X, \tau) = 4 - \frac{T_{k+1} - \tau}{\tan \phi} = 4 - \rho, \text{ where } \rho = \frac{\tau}{\tan \phi}.$$

$\square$

We observe that in both cases in the analysis of Theorem D.1 we obtain that $f \in (2, 4]$, as a function of $\tau$ and $\phi$. This result makes intuitively sense, since $X$ is 4-robust, and the smallest consistency is equal to 2 (when $\phi \to 0$).

# E   Further experimental analysis

To further quantify the performance difference between the two algorithms, PROFILE and PO, we performed additional experiments. Specifically, we used a list of the last 20,000 minute-exchange rates of BTC to USD, so as to create 20 different sequences, each with its own prediction, using the same method as in Fig 2c. For each sequence, we computed the average improvement over PO for rates in the interval of interest $[0.9\hat{p}, 1.1\hat{p}]$. Figure 7 depicts this average for each of the 20 sequences. We observe that for the sequences in which PROFILE outperforms PO (12 out of 20), the improvement ranges from roughly $15\%$ to $30\%$, whereas PO outperforms PROFILE in 8 out of 20 sequences, by a factor that is at most $10\%$, roughly.

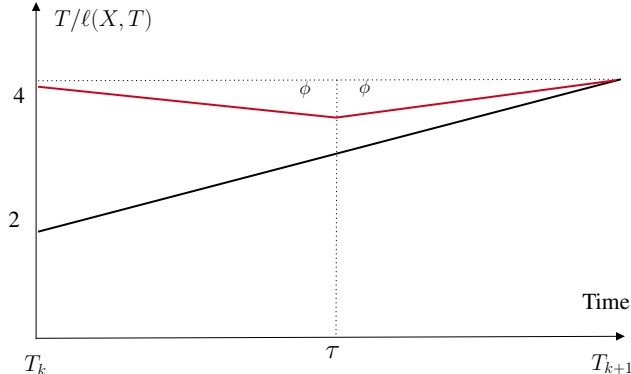

Figure 6: An illustration of Case 2.

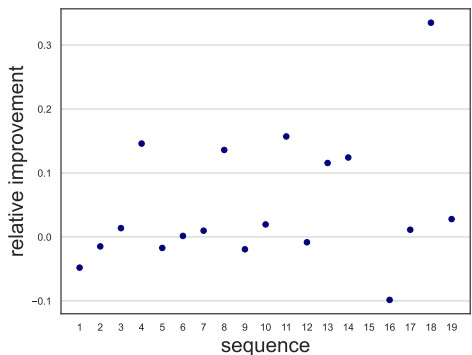

Figure 7: Average ratio improvement of PROFILE over PO

# F   Computational setup

The experiments are reproducible on any standard computer, and do not require any memory or computational power beyond the standard requirements. They run typically within few milliseconds.

