# OpenReview forum: "Overcoming Brittleness in Pareto-Optimal Learning Augmented Algorithms"
_NeurIPS.cc/2024/Conference — NeurIPS 2024 poster_

### Official Review · Reviewer_czee · 2024-07-08

**Soundness:** 3
**Presentation:** 3
**Contribution:** 3
**Rating:** 6
**Confidence:** 4

**Summary:**

In the burgeoning field of learning-augmented online algorithms, the ideal case is to design an algorithm that achieves a competitive ratio (CR) as a function of prediction errors, without knowing the prediction error in advance. To tackle this problem, many existing works focus on two extreme metrics: consistency (i.e., the CR when the prediction error is zero) and robustness (i.e., the worst CR over all prediction errors). These works aim to derive algorithms that achieve the Pareto-optimal trade-off between consistency and robustness.

This paper introduces and studies the concept of brittleness in Pareto-optimal learning-augmented algorithms, highlighting that the CR of Pareto-optimal algorithms may sharply degrade to the robustness guarantee with only a small prediction error. Specifically, the work formally defines and demonstrates the brittleness of the max-rate prediction in the one-way trading problem. To overcome this brittleness, it extends the consistency to a concept of consistency by profile F, which specifies the target CR over different prediction errors. A profile-based algorithm is proposed to check the feasibility of the consistency by profile F and find the online algorithm if feasible algorithms exist to attain it.

Finally, the paper presents an adaptive algorithm that can improve the performance of Pareto-optimal algorithms for particular instances. However, this algorithm also suffers from brittleness.

**Strengths:**

- This paper formalizes the brittleness issues in Pareto-optimal algorithms for the one-way trading problem. This helps to better understand the limitations of existing works in learning-augmented algorithms, which is particularly important for practical applications where perfect predictions are nearly impossible.

- The concept of consistency by profile is a natural yet useful extension of the classic consistency. The proposed algorithm, which can quickly identify feasibility and find a feasible online strategy, is an interesting extension of classic threshold algorithms.

**Weaknesses:**

-  The paper focuses solely on the brittleness issues for the one-way trading problem. It is unclear whether many other problems exhibit similar brittleness and whether profile-based algorithms can also be designed to address such brittleness. The paper mentions the contract scheduling problem in the appendix, which deserves more formal treatment. In contrast, the paper uses an entire section (Section 5) to propose an adaptive Pareto-optimal algorithm (which is still brittle) for one-way trading, which deviates from the central topic of overcoming brittleness in learning-augmented algorithms.

- The concept of consistency by profile is relatively easier to define for single-value predictions, raising the question of whether this concept would generalize when considering multiple predictions. Specifying the profile may also be challenging for users.


- The algorithmic techniques used to design profile-based algorithms seem similar to existing approaches, which involve designing thresholds to maintain the target consistency profile while being prepared for the worst-case scenario if exchange rates drop to 1. The complexity arises from maintaining consistency over the user-specified profile instead of a single perfect prediction point.

**Questions:**

- Can you explain how Section 5 (adaptive brittle algorithm for one-way trading) is connected to the theme of the paper (overcoming brittleness in learning-augmented algorithms)? How significant are the results for contract scheduling in the appendix, and can these results be used to validate the generalizability of the proposed concepts and approaches?

- For a given user profile, there may exist multiple algorithms that can ensure such a profile; however, these algorithms may exhibit different instance-dependent performances. Can you make some formal statements on how to select the profile-based algorithm in practice? It seems the algorithm making $\omega_{l+1} = 1$ can be a good candidate algorithm.

- The definition of brittleness (in Definition 3.1) is specific to the maximum rate prediction of one-way trading. Can this be a more general definition for the brittleness of learning-augmented algorithms?

- in line 216 page 5, should the integral in $\tilde{s}_i = \int_1^{w_i}\Phi(u)du$ start from $0$?

**Limitations:**

Yes

---

> ### Author Rebuttal · Authors · 2024-08-04
>
> Thank you for your feedback. We comment first on “weaknesses”.
>
> **1**. There are indeed other problems for which Pareto-optimal algorithms are brittle. Examples include 1-MAX search [19], [38] which is a much simpler version of one-way trading, online bidding [6], [23] and searching for a hidden target in the infinite line [4]. We believe there may be several other problems in this class; for example, given the connections between one-way trading and online knapsack demonstrated in [16], it is quite likely that knapsack also suffers from brittleness, in the maximum-density prediction setting studied very recently in [A], but this remains to be proven. More broadly, we aim to bring attention to the fact that Pareto-optimality treats the prediction error in an “all-or-nothing” fashion (perfect vs adversarial predictions) which may not always yield an appropriate measure of performance. Concerning the adaptive algorithm please see our response to Question 1 below.
>
> **2**. In regards to applying the model to settings beyond single-valued predictions, please refer to *Point 2* in the *global response*. As we explain, while the model is indeed more amenable to single predictions, it can be applied to more complex prediction settings. Specifying the user-based profile can indeed be challenging for certain problems. But we also believe that for problems such as one-way trading, and online financial optimization problems, more broadly, a user-based profile makes sense: e.g., a trader may be satisfied with a linear-like degradation of performance, based on historical data from stock exchanges.
>
> **3**. Designing a threshold function for the profile setting is non-trivial, and does not follow straightforwardly from known approaches. Please refer to *Point 3* in our *global response*.
>
> Below we respond to questions:
>
> **1**. The adaptive setting of Section 5 stems from observing that the design and analysis of SOTA algorithms are heavily tied to worst-case sequences, yet a natural question is to ask: can we design Pareto-optimal algorithms that improve upon the SOTA if the input diverges from the worst-case one? We believe this is a natural question that falls into the *analysis beyond the worst-case*. It is also related to your observation about the analysis being tied to worst-case sequences: we show that it does not need to, and we can indeed go beyond. To solve this problem, we applied some techniques we developed for the profile setting: namely, when analyzing constraints $[\beta]$, we argue that replacing the inequality with an equality allows us to analytically solve the fundamental differential equation, which differs from previous analysis techniques of threshold algorithms. Thus, we believe this setting is not disconnected from the profile setting of Section 4. We also believe that one can combine the two approaches, and obtain an adaptive, profile-based algorithm, in order to circumvent the unavoidable brittleness. This is done as follows, in rough terms: For the decreasing part of the profile (rates smaller than the prediction), the algorithm behaves similarly to lines 6-9 of Alg. 2, by replacing $r$ with the individual ratios $t_i$. For the increasing part of the profile, we would like to exchange as much as possible at each rate, subject to the profile. This can be accomplished, at a high level, using an approach along the lines of Appendix C, but with some additional technical modifications due to the presence of multiple ratios $t_i$ instead of a single one.
>
> In regards to contract scheduling, we demonstrate how to analytically find a schedule that simultaneously optimizes the robustness and the consistency according to a given linear profile. More precisely, we present a 4-robust schedule (best-possible) that also has optimal consistency according to this profile. This demonstrates that the model applies to other problems. Contract scheduling is an important problem in AI that has been studied in learning-augmented settings [7],[B]. Moreover, it has clear connections to other problems such as online-bidding [6,23] and searching on the line [4]. We are very confident that our approach will carry over to these problems, which, likewise, suffer from brittleness.
>
> **2**. You are correct in that there may exist several algorithms that respect a given profile, say $F$,  and one would like to define further criteria to choose a good one. One way to accomplish this is to use our offline algorithm so as to find the best-possible extension $G$ of $F$, as stated in Remark 4.1, and in the discussion at the end of Section 3, starting at line 185. Intuitively, this extension $G$ describes a profile that has the same “shape” as $F$, but defines much better performance ratios than $F$, for all rate values. This means that if $F$ is feasible, then we can obtain an algorithm that not only respects $F$, but also $G$ (hence will perform even better, and “optimally” in the sense of respecting the "lowest" profile that has the shape of $F$). One could impose other criteria, e.g., insist that   $w_{l+1} =1$ as you suggest.
>
> **3**. Yes, the definition can be extended as follows. Let $O$ be an online problem (say cost minimization). Let $\hat{p}$ denote a prediction, $\eta$ the metric that defines the prediction error, and let $r$ denote the robustness requirement. We say that $O$ is *brittle with respect to $\hat{p}$* if for every Pareto-optimal algorithm PO and every $\epsilon > 0$, there exists a sequence $\sigma$ such that $\eta(\sigma,\hat{p}) \leq \epsilon$, and ${cost}(PO,\sigma)\geq r \cdot cost(OPT,\sigma) -\delta$, where $\delta$ can be infinitesimally small.
>
> **4**. Correct, thank you for catching the typo.
>
> [A] M. Danashveramoli et al: Competitive Algorithms for Online Knapsack with Succinct Predictions, arXiv:2406.18752
>
> [B] S. Angelopoulos et al. “Contract Scheduling with Distributional and Multiple Advice”, arXiv:2404.12485

---

> > ### Comment · Reviewer_czee · 2024-08-09
> > **Response to rebuttal**
> >
> > Thank you for your reply. I still find that Section 5 is disconnected from the core theme related to overcoming the brittleness of learning-augmented algorithms, unless this section can formally address the brittleness of the adaptive algorithm, as the authors believe it can.
> >
> > After further consideration of the paper's potential impact, I believe it makes a valuable contribution by bringing the issue of brittleness in Pareto-optimal algorithms to the attention of the field. Therefore, I have increased my score from 5 to 6.

---

### Official Review · Reviewer_9nZL · 2024-07-09

**Soundness:** 3
**Presentation:** 3
**Contribution:** 3
**Rating:** 6
**Confidence:** 3

**Summary:**

The authors consider learning-augmented algorithms for the one-way trading problem. In this problem, we are given a budget of 1 and a sequence of exchanges rates between 1 and M that are revealed in an online manner. Whenever an exchange rate is revealed, we have to decide whether to exchange a fraction of our remaining budget at this rate or not. The goal is to maximize the overall profit. In the learning-augmented setting, we are additionally given a prediction on the maximum exchange rate, which is equivalent to predicting the optimum, as an optimal solution will exchange the complete budget at the maximum rate. Usually, learning-augmented algorithms are analyzed using consistency, the competitive ratio for a perfectly accurate prediction, and robustness, the worst-case competitive ratio for any input. For the one-way trading problem, a pareto optimal algorithm w.r.t. consistency and robustness is already known from previous works.

The main point of the paper is to address two weaknesses of such pareto-optimal algorithms and analyses via consistency and robustness in general. The first weakness is called *brittleness* and describes prediction models were even a wrong prediction that is arbitrarily close to the correct value leads to any pareto-optimal algorithm having a competitive ratio matching the robustness. This means that only completely perfect predictions allow for an improved performance. The authors prove that maximum rate predictions for the one-way trading problem are brittle and address the brittleness by proposing an analysis via *profiles*, a generalization of consistency and robustness. A $profile$ partitions the range of possible maximum exchange rates into intervals and, for each interval, defines a target competitive ratio. The interval that contains the predicted maximum exchange rate has the best target competitive ratio and the ratio degrades for intervals that are farther away. An algorithm respects the profile if it always achieves the target competitive ratio of the interval that contains the actual maximum exchange rate. As a main contribution, the paper gives a constructive algorithm that decides whether there exists an algorithm that respects a given profile.

The second weakness of pareto-optimal algorithms is that such algorithms are often tailored to worst-case instances w.r.t. the consistency and robustness tradeoff. To address this, the authors give a pareto-optimal algorithm that dominates all other pareto-optimal algorithms in the sense that it does not perform worse than any other pareto-optimal algorithm on instances where the actual maximum exchange rate is larger than the predicted optimal exchange rate.

**Strengths:**

* The paper identifies and addresses two reasonable and realistic potential drawbacks of pareto-optimal learning-augmented algorithms. Since the framework of learning-augmented algorithms is ultimately a tool to analyze algorithms beyond the worst-case, the proposed generalizations that extend the ability to do just that are certainly of interest to the community.

* The authors give a proof of concept that profile-based algorithms are indeed possible by introducing such algorithms for the one-way trading problem (and contract scheduling in the appendix). It is a nice idea to consider the offline problem of deciding whether a given profile is feasible or not. I am not completely convinced that this idea also works for problems of a different flavor (see weaknesses), but at the very least it inspires future work to settle this question, which already is an important contribution.

* The paper is well-written, the problem statement is properly motivated, and the results are presented clearly and concisely.

**Weaknesses:**

* The definition of profiles seems to be tailored to predictions models where only a single value is predicted. For more complex predictions, one could define profiles w.r.t. the prediction error. However, specifying a profile upfront would not always be possible as the range for the predictions error often depends on the unknown online input. Even for the maximum exchange rate predictions, the specification of a profile requires knowledge of the maximum exchange rate M.

* As one of the main technical contributions, the authors give an algorithm that decides whether a given profile is feasible. This algorithm heavily relies on the simple characterization of worst-case instances as given in Remark 2.1. For many other online problems, no such simple characterization of worst-case instances are known. This could limit the impact of the proposed analysis framework via profiles as deciding whether a profile is feasible is likely more difficult (or even not possible) for problems without such a simple worst-case characterization.

* I am not an expert regarding the literature on the one-way exchange problem. However, the paper does not seem to introduce too many new algorithmic ideas. Instead, the algorithmic contributions of the paper seem like a very natural extension of known ideas. In particular, Algorithm 1 seems to be the canonical way of extending threshold-based algorithms to profiles.

* Regarding the experiments, it would be interesting to also see results for different profiles. The used profile is quite similar to the consistency and robustness cases. While this makes the comparison to the pareto-optimal algorithm more fair, it would be interesting to see results for profiles that emulate a smooth error dependency.

* Minor comment: Line 86: I do not think that not having access to the prediction ahead of time is a novelty. In the context of online algorithms there are several examples where the predictions also are revealed over time. One example would be reference [9].

**Questions:**

-

**Limitations:**

In my opinion, all limitations have been properly addressed.

---

> ### Author Rebuttal · Authors · 2024-08-04
>
> Thank you for your feedback. Below we respond to “weaknesses”.
>
> **1**.  In regards to the prediction being tied to single values, please see *Point 2* in the *global response*. As we explain, while the model is indeed more amenable to single predictions, it can apply to more complex prediction settings.
>
> In regards to the error, specifically, there is a variety of problems and prediction settings for which the worst-case prediction error is bounded, with or without any further assumptions. An example where no additional assumptions are needed includes online problems with frequency-type predictions, e.g., bin packing  [8], or knapsack [A]. In our problem, the prediction error is bounded from above by $M$, by the assumption that $M$ is the maximum exchange rate. But this is a standard assumption in the context of trading problems, not only in the standard competitive analysis of one-way trading [19] (without this assumption, no algorithm has bounded competitive ratio), but also in the state-of-the-art learning-augmented algorithm [38]. The bound helps compare our algorithm to that of [38]; however, it is not strictly needed in the definition of the profile, since the profile can be defined even if the error is unbounded. Hence, having an upper bound on the prediction error may be helpful, but is not a requirement in our model.
>
> [A] Im, Sungjin, et al. "Online knapsack with frequency predictions." NeuRIPS (2021): 2733-2743.
>
> **2**. In general, the analysis of profile-based algorithms need not rely on knowing the structure of worst-case sequences, in the same way that competitive analysis in the standard setting (without any predictions) need not rely on such knowledge. Knowing this structure may help the analysis, but is not a prerequisite. For instance, our analysis of contract scheduling (appendix) does not use “worst-case” instances, it applies instead to any given instance. In addition, we believe that one-way trading remains a challenging problem in our setting even when knowing the structure of worst-case instances, because there are several conflicting objectives in trade-off relation, which is reflected in the complexity of the algorithms and their analysis.
>
> **3**. We address this issue in *Point 3* of the *global response*, which we also include below. Designing a threshold function for the profile setting is non-trivial, and does not follow straightforwardly from known approaches. For instance, if one tried a “myopic” approach that considered each interval individually (and obtained a threshold function for each such interval), then the overall function would fail. This is because when transitioning to a new interval, the algorithm would not have made enough profit to be competitive in this new interval. This adds complications which we address as explained in Section 4 (lines 245-254) and in lines 9-14 of Algorithm 1: informally, we need to “flatten” a portion of the threshold function of each interval appropriately. As a result, the obtained function is quite complex, and combines exponential functions, plateaus and discontinuities, as illustrated in Figure 3 (appendix).
>
> **4**. For the experiments, we chose a relatively simple profile in order to be able to compare our algorithms to the known Pareto-optimal algorithms in a very clear and meaningful manner . However, we agree with your suggestion, and in *Point 1* of the *global response* and the accompanying PDF,  we describe the performance of our algorithm on a more complex profile, which demonstrates that the algorithm indeed performs as predicted by the theoretical results.
>
> **5**. Here, we meant that this is a novelty in regards to one-way trading, and online financial optimization problems, more broadly. We will clarify this point, and add references to [9] and other related works.

---

> > ### Comment · Reviewer_9nZL · 2024-08-09
> > **Response to rebuttal**
> >
> > Thank you for your rebuttal on my review and the other reviews. My opinion on the paper and my score remain unchanged.

---

### Official Review · Reviewer_jsKn · 2024-07-13

**Soundness:** 3
**Presentation:** 2
**Contribution:** 3
**Rating:** 6
**Confidence:** 4

**Summary:**

The paper considers the learning-augmented one-way trading problem. In the problem, we are given a starting budget equal to 1 and a sequence of exchange rates $p_1,...,p_n \in [1,M]$ arriving online. When each $p_i$ arrives, we need to. decide the amount to be exchanged to the secondary currency. Our goal is to maximize the total profit under the given budget. In the learning-augmented setting, the algorithm can access an imperfect prediction $\hat{p}$ of the largest rate.

The authors first show that Pareto-optimality is very fragile for comparing online trading algorithms and then motivates a new metric called performance profile. This new metric incorporates the structural information of instances, rather than a simple comparison based on consistency-robustness values. They further develop an online algorithm that can satisfy the given performance profile (if it is feasible). The authors also discuss another generalization of  Pareto-optimality and provide empirical evaluations of the proposed algorithms in the end.

**Strengths:**

- The brittleness issue considered in the paper is well-motivated. Actually, this is a highly significant concern in the field of learning-augmented algorithms. Previous efforts mainly focused on achieving smoothness in ratios by defining new error metrics, while this paper takes a novel approach by introducing the concept of profiles to tackle this issue.

- Both theoretical analysis and experimental evaluation are provided in the paper.

**Weaknesses:**

- The main weakness of this work is that the concept of performance profile may be hard to extend to other online problems, which makes this work less interesting. It would be better if the authors could demonstrate the applicability of this technique to a wider range of problems.

- In Line 131, I didn't see why $w_{A,i}=\sum_{j=1}^{i-1}w_{A,j}$. Is this a typo?

**Questions:**

See the weakness above.

**Limitations:**

I didn't see any potential negative societal impact.

---

> ### Author Rebuttal · Authors · 2024-08-04
>
> Thank you for your feedback. In regards to weaknesses/questions, our responses are below.
>
> **1**. We address this issue in *Point 2* of the *global response*, which we also include below for convenience.
>
> The concept of a profile is inherently applicable, and at the very least, to the class of online problems with a single-valued prediction. In this work, we focused on two well-known problems from this class. The first and main problem is one-way trading, which was chosen because it is one of the main online financial optimization problems and an error-based analysis for such problems is obviously paramount (but missing in the state of the art). In addition, the problem has very close connections to other important online problems, notably online knapsack [16], [41]. The second application is contract scheduling, because it is a well-known problem from AI with close connections to other important problems such as online bidding [6], [23], as well as searching for a hidden target [4], both of which suffer from brittleness. The concepts and techniques we introduced should be readily applicable to such related problems, but due to space limitations and the complexity of approaches, we had to make a selection.
>
> There are two additional points we would like to further emphasize. The first is that single-valued predictions constitute a very rich class of learning-augmented algorithms.  Beyond the works cited above, many other studies fall in this class including ski rental and rent-or-buy problems [36], [39], scheduling [A], secretary problems [10] and bin packing [A], to mention only a few representative works. Such predictions are also very useful in the context of *succinct* predictions, e.g., as studied explicitly in the recent work [D].
>
> The second point is that the prediction need not necessarily be single-valued for our profile-based analysis to be applicable. For example, the prediction may be a *vector* of values, as e.g., in scheduling [26] or bin packing [8], then the concept of profile still applies since the error is defined by a distance norm between the predicted and the actual vector. Our model can also be applicable in a *multiple* prediction setting, in which the algorithm is given a set of several predictions, and its consistency is evaluated at the best-possible prediction in this set. More concretely, we believe that one could combine our analysis of one-way trading with the multiple advice setting of [B], and our analysis of contract scheduling with the multiple advice setting of [C]. Of course, we expect any technical results to be more challenging.
>
> Nevertheless, we agree that our profile model, as is, is not immediately applicable to all learning-augmented settings, e.g., when predictions appear dynamically. This is a topic of future work, and we will emphasize this in the introduction and the conclusions.
>
> [A] K. Anand et al. "A regression approach to learning-augmented online algorithms." NeuRIPS 34 (2021): 30504-30517.
>
> [B] K. Anand et al. "Online algorithms with multiple predictions". ICML 2022, 582-598.
>
> [C] S. Angelopoulos et al. “Contract Scheduling with Distributional and Multiple Advice”, arXiv:2404.12485
>
> [D] M. Danashveramoli et al: "Competitive Algorithms for Online Knapsack with Succinct Predictions", arXiv:2406.18752
>
>
> **2**. Yes this is a typo. The correct expression is  $w_{A,i}=w_{A,i-1} + x_i$, where $x_i$ is the amount traded on the $i$-th rate. I.e., $w_{A,i}$  is the sum of the amounts exchanged up to and including the $i$-th request.

---

> > ### Comment · Reviewer_jsKn · 2024-08-11
> >
> > Thank the authors for the comments. I will maintain my original score.

---

### Official Review · Reviewer_f8Kk · 2024-07-13

**Soundness:** 4
**Presentation:** 3
**Contribution:** 3
**Rating:** 8
**Confidence:** 3

**Summary:**

In the context of learning augmented algorithms, two widely used metrics are robustness (i.e: the performance when the prediction is adversarially chosen) and consistency (i.e: the performance when the prediction is perfect).

This work analyzes the interplay between these two metrics in the one way trading problem with imperfect predictions. In particular, it starts by showing that Pareto-optimal algorithms, that is, algorithms that for a given robustness upper bound r achieve the smallest possible consistency c(r), are dramatically sensitive to small prediction errors. Specifically, the authors show that any prediction error guarantees the existence of an input sequence for which the performance ratio reaches its robustness, a behavior denominated ‘britleness’. This implies that the competitive ratio of any Pareto-Optimal algorithm is either its consistency (if the prediction is indeed perfect) or its worse possible performance, suggesting that Pareto-Optimality might not be a sensible algorithm design criteria.

In light of this result, this work puts forward the notion of profiles, which maps rate prediction intervals to desired competitive ratios, allowing for the performance of algorithms to degrade smoothly with respect to prediction errors. *V*iewing this, the a*u*thors present an a*l*gorithm that establishes the *f*easibility of a profile in an offline fashion and yields an online procedure that, in the feasible case, satisfies the enforced performance constraints.

Lastly, the authors propose an adaptive algorithm that is not designed to handle worst-case predictions but leverages the deviations from the predicted exchange rate to navigate the robustness and consistency tradeoff.

**Strengths:**

- The paper is clearly written.

- The theoretical results (brittleness of Pareto-Optimality,  feasibility determination of profile-following and correctness of online algorithm for profile following) are relevant and sound.

- The analysis technique used to determine the feasibility and solve the profile-based one-way trading problem seems novel. In particular, the constraints associated to a profile can be written as a set of linear differential differential equation whose solution yields a profile-following exchange strategy.

- The authors provide intuition regarding the PROFILE algorithm, in particular, on the behavior of the threshold function $\phi$ with respect to transitions in the desired performance ratio.

**Weaknesses:**

- The experimental setting might be limited. Specifically, the profile used for evaluation if fairly simple, with only three intervals, two of which map to the worse possible ratio.

- Is there any practical use of determining feasibility if it's in an offline fashion ?

**Questions:**

- Is there a way to characterize the likelihood of the sequences that severely degrade the performance of Pareto-Optimal algorithms ? If those sequences are very unlikely then Pareto-Optimality might not be that fragile a criteria.

- Why is the function \phi increasing ? That is, why do larger utilizations necessarily map to larger reservation rates ? (Is it only under the assumption that rates are increasing and then drop to 1?)

Minor comment: Fix figure ratios.

**Limitations:**

- N/A

---

> ### Author Rebuttal · Authors · 2024-08-04
>
> Thank you for your feedback. Please allow us first to comment on the “weaknesses”.
>
> **1**. For the experiments, we chose a relatively simple profile in order to be able to compare our algorithms to the known Pareto-optimal one, in a very clear and meaningful manner. Nevertheless, we agree with your suggestion, and in *Point 1* of the *global response* and the accompanying PDF, we describe the performance of our algorithm on a more complex profile, which demonstrates that the algorithm indeed performs as predicted by the theoretical results even on more complex profiles.
>
> **2**. There are indeed practical uses for determining feasibility in an offline fashion. Specifically, let $F$ denote the given profile, then we can use the offline algorithm combined with binary search, so as to find the best-possible extension $G$ of $F$, as stated in Remark 4.1, and in the discussion at the end of Section 3, starting at line 185. Intuitively, this extension $G$ describes a profile that has the same overall “shape” as $F$, but defines much better performance ratios than $F$ for all rate values. This means that if $F$ is feasible, then we can obtain an online algorithm that not only respects $F$, but also $G$ (hence will perform even better).
>
> Below, please see our response to the questions:
>
> **1**. We are not aware of any such characterizations in the literature, not only for Pareto-optimal algorithms, but also for the standard competitive analysis of this problem. Our adaptive algorithm of Section 5 aims to address this concern: if the sequence is not pathological, then we show that we can perform much better than the known Pareto-optimal algorithms (while maintaining Pareto-optimality). But nothing formal is known about the “likelihood” that a sequence severely degrades the performance of such algorithms. An additional complication is that Pareto-optimality is a generalization of competitive analysis, and thus it is intrinsically bound to worst-case analysis.
>
> **2**. This is due to Remark 2.1:  Any optimal algorithm only trades at rates which are local maxima, hence the function $\phi$ must be increasing for the algorithm to be optimal. Furthermore, $\phi$ needs to be invertible, in order to determine utilization and thus the exchanges made at each rate, hence it needs to be increasing.
>
> Thank you for the suggestion, it is not immediately clear to us to which figure you refer, and whether you mean that certain figure ratios should be more legible, but we will do so.

---

> > ### Comment · Reviewer_f8Kk · 2024-08-09
> >
> > The authors have addressed my questions, and I have raised my score to 8.

---

### Author Rebuttal · Authors · 2024-08-04

We respond to some points brought up in the reviews.

**1**. **Experimental evaluation on complex profiles**.  For the experiments, we chose a relatively simple profile in order to be able to compare our algorithms to the known Pareto-optimal one, in a very clear and meaningful manner. Namely, the profile of Fig 2(a) in the submission captures the consistency/robustness tradeoff, the smoothness around the prediction, but also allows for an average-improvement evaluation as in Fig. 2(c), which becomes a much more subjective task for more complex profiles. Nevertheless, we agree with the suggestion of some reviewers, and in the accompanying PDF we consider a more complex profile, shown in Fig. 1. Fig. 2 in the PDF depicts the performance of PROFILE relative to the SOTA Pareto-Optimal algorithm PO. We observe again that PO has high brittleness if $p^*$ is close to, but smaller than $\hat{p}$, whereas PROFILE has a much smoother overall performance that respects the profile of Fig. 1 and again validates Theorem 3.1.

**2**. **Prediction types**. The concept of a profile is inherently applicable, and at the very least, to the class of online problems with a single-valued prediction. In this work, we focused on two well-known problems from this class. The first and main problem is one-way trading, which was chosen because it is one of the main online financial optimization problems and an error-based analysis for such problems is obviously paramount (but missing in the state of the art). In addition, the problem has very close connections to other important online problems, notably online knapsack [16], [41]. The second application is contract scheduling, because it is a well-known problem from AI with close connections to other important problems such as online bidding [6], [23], as well as searching for a hidden target [4], both of which suffer from brittleness. The concepts and techniques we introduced should be readily applicable to such related problems, but due to space limitations and the complexity of approaches, we had to make a selection.

There are two additional points we would like to further emphasize. The first is that single-valued predictions constitute a very rich class of learning-augmented algorithms.  Beyond the works cited above, many other studies fall in this class including ski rental and rent-or-buy problems [36], [39], scheduling [A], secretary problems [10] and bin packing [A], to mention only a few representative works. Such predictions are also very useful in the context of *succinct* predictions, e.g., as studied explicitly in the recent work [D].

The second point is that the prediction need not necessarily be single-valued for our profile-based analysis to be applicable. For example, the prediction may be a *vector* of values, as e.g., in scheduling [26] or bin packing [8], then the concept of profile still applies since the error is defined by a distance norm between the predicted and the actual vector. Our model can also be applicable in a *multiple* prediction setting, in which the algorithm is given a set of several predictions, and its consistency is evaluated at the best-possible prediction in this set. More concretely, we believe that one could combine our analysis of one-way trading with a multiple advice setting, such as the one studied in [B] (with static predictions), and our analysis of contract scheduling with the multiple advice setting of [C]. Of course, we expect any technical results to be more challenging.

Nevertheless, we agree that our profile model, as is, is not immediately applicable to all learning-augmented settings, e.g., when predictions appear dynamically. This is a topic of future work, and we will emphasize this in the introduction and the conclusions.

[A] K. Anand et al. "A regression approach to learning-augmented online algorithms." NeuRIPS 34 (2021): 30504-30517.

[B] K. Anand et al. "Online algorithms with multiple predictions". ICML 2022, 582-598.

[C] S. Angelopoulos et al. “Contract Scheduling with Distributional and Multiple Advice”, arXiv:2404.12485

[D] M. Danashveramoli et al: "Competitive Algorithms for Online Knapsack with Succinct Predictions", arXiv:2406.18752


**3**. **Challenges in the design and analysis**. Designing a threshold function for the profile setting is non-trivial, and does not follow straightforwardly from known approaches. For instance, if one tried a “myopic” approach that considered each interval individually (and obtained a threshold function for each such interval), then the overall function would fail. This is because when transitioning to a new interval, the algorithm would not have made enough profit to be competitive in this new interval. This adds complications which we address as explained in Section 4 (lines 245-254) and in lines 9-14 of Algorithm 1: informally, we need to “flatten” the threshold function of each interval appropriately. As a result, the obtained function is quite complex, and combines exponential functions, plateaus and discontinuities, as illustrated in Figure 3 (appendix).

---

### Decision · Program_Chairs · 2024-09-25

**Decision:**

Accept (poster)

**Comment:**

This manuscript identifies and studies the brittleness problem in the usual Pareto-optimal learning algorithms, through the example of one-way trading. Specifically, it is shown that all Pareto-optimal algorithms (i.e. achieving the best competitive ratio with a perfect prediction, while maintaining a prescribed worst-case competitive ratio for possibly bad predictions) are brittle in the sense that the competitive ratio becomes the worst even if the prediction is very accurate (but not perfect). Two approaches are proposed to remedy this brittleness: 1) a profile-based approach is proposed to allow for different competitive ratios on different intervals, and an efficient algorithm is developed to tell if any given profile is achievable; 2) an adaptive Pareto-optimal algorithm is developed whose performance depends on the distance to the worst input sequence.

The identification of the brittleness of Pareto-optimal algorithms is significant in practical economic applications. The proposed ideas to remedy this drawback are also sound, and practically applicable supported by the numerical experiments. This manuscript is unanimously appreciated by the reviewers, and I'll happily recommend acceptance.